# New Azo Dyes-Based Mg Complex Pigments for Optimizing the Anti-Corrosion Efficiency of Zinc-Pigmented Epoxy Ester Organic Coatings

Miroslav Kohl [1,*], Fouzy Alafid [1], Karolína Boštíková [1], Marek Bouška [1], Anna Krejčová [1], Jan Svoboda [1], Stanislav Slang [1], Ludmila Michalíčková [1], Andréa Kalendová [1], Radim Hrdina [1], Ladislav Burgert [1], Eva Schmidová [2], Pravin P. Deshpande [3] and Abhijit A. Bhopale [3]

1   Faculty of Chemical Technology, University of Pardubice, Studentská 573, 532 10 Pardubice, Czech Republic; fawzi_r@yahoo.com (F.A.); karolina.bostikova@student.upce.cz (K.B.); marek.bouska@upce.cz (M.B.); anna.krejcova@upce.cz (A.K.); jan.svoboda@upce.cz (J.S.); stanislav.slang@upce.cz (S.S.); ludmila.michalickova@student.upce.cz (L.M.); andrea.kalendova@upce.cz (A.K.); radim.hrdina@upce.cz (R.H.); ladislav.burgert@upce.cz (L.B.)
2   Jan Perner Transport Faculty, Educational and Research Centre in Transport, University of Pardubice, Doubravice 41, 533 53 Pardubice, Czech Republic; eva.schmidova@upce.cz
3   Department of Metallurgy and Material Science, College of Engineering Pune, Pune 411005, India; pravinpd@hotmail.com (P.P.D.); bhoplea.meta@coeptech.ac.in (A.A.B.)
*   Correspondence: miroslav.kohl@upce.cz; Tel.: +420-446-037-272

**Abstract:** This work addresses the possibilities of using synthesized novel magnesium complex dyes in zinc pigmented organic coatings based on epoxyester resin to reduce the zinc content in these coatings while maintaining or increasing the anticorrosive efficiency of them. The magnesium complexes Mg-Dye-I ($C_{34}H_{26}MgN_8O_6$), Mg-Dye-II ($C_{26}H_{19}MgN_3O_5$), Mg-Dye-III ($C_{17}H_{10}MgN_2O_3$), and Mg-Dye-IV ($C_{25}H_{18}MgN_4O_6$) with a series of azo carboxylate ligands were prepared from the diazo-coupling reaction of anthranilic acid with 5-methyl-2-phenyl-3-pyrazolone (Dye I; $C_{17}H_{14}N_4O_3$), anthranilic acid with naphthol AS-PH (Dye II; $C_{26}H_{21}N_3O_5$), anthranilic acid with 2-naphthol (Dye III; $C_{17}H_{12}N_2O_3$), and 2-amino-5-nitrophenol with naphthol AS-PH (Dye IV; $C_{25}H_{20}N_4O_6$). The synthesized novel magnesium complex dyes were characterized by analytical methods. Model coatings containing these dyes at pigment volume concentrations (PVCs) = 1, 3, 5 and 10% and zinc at a ratio of pigment volume concentration/critical pigment volume concentration (PVC/CPVC) = 0.60 were formulated to study the inhibitory properties of the individual synthesized magnesium complex dyes. Model coatings containing inorganic pigments (MgO and Ca-Mg-HPO$_4$) at PVCs = 1%, 3%, 5% and 10% and zinc at PVC/CPVC = 0.60 were also formulated. The coating pigmented only by zinc at PVC/CPVC = 0.60 was prepared as a standard organic coating. Corrosion resistance was also evaluated by potentiodynamic polarization studies and electrochemical impedance spectroscopy. The properties of organic coatings were also tested using other standardized and derived corrosion tests. In addition, the mechanical properties of the studied organic coatings were determined using standard tests. The aim of the work was to verify the possible synergistic efficiency of novel magnesium complex dyes by improving the mechanical, anti-corrosion, and chemical properties of zinc pigmented organic coatings.

**Keywords:** magnesium complex; azo dyes; coating; corrosion; anticorrosion efficiency

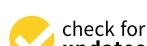



## 1. Introduction

Corrosion of carbon steel not only exerts a significant economic burden but also impacts environmental pollution and the safety of human life [1–3]. The most widespread method of protecting metal materials is represented by organic coatings, the anti-corrosion effectiveness of which depends mainly on the pre-treatment of the metal surface; the concentration, solubility, and type of the anti-corrosion pigment; the method of creating

the coating film; the adhesion of the coating to the substrate; the type of binder; and the mechanical properties of the entire coating system [4–7].

The most common variant for mitigating the effects of a corrosive environment on the material and thereby minimizing economic losses is the use of corrosion inhibitors. Corrosion inhibitors are compounds introduced in small amounts into the working liquid that adsorb on the surface of the metal and act as a barrier between the protected metal and the corrosive environment [8,9]. In the past, the most used inhibitory pigment was zinc chromate, which is slightly soluble in water and, in this way, releases inhibitory substances that adsorb on the active sites of the metal and protect it from corrosion through a film formation mechanism. However, the use of zinc chromate in protective coatings today has been drastically reduced due to its toxicity, and research is focused on finding an environmentally safer and less toxic alternative [10,11]. It is currently difficult to achieve high anti-corrosion efficiency and low cost, which are required when applied to a corrosive environment, with a single non-toxic corrosion inhibitor. However, it has been proven that, by mixing various non-toxic corrosion inhibitors, their synergistic effect can be used, and an anti-corrosion pigment with high anti-corrosion efficiency can be obtained [12]. Currently, the commonly used environmentally friendly corrosion inhibitors include organic compounds containing heteroatoms and $\pi$ electrons [13–15], for example, imidazoline [16–18], thia-diazole derivatives [19–21], and amino acid [22–24] or sulfhydryl compounds [25].

Zinc-filled organic coatings have been used for decades to protect steel structures from corrosion. In zinc-rich coatings, electrochemically active zinc particles act as anodes and are sacrificed to provide cathodic protection to the steel substrate. Due to the potential difference, a Zn–Fe corrosion cell is formed. Corrosion occurs at places with more negative potential (anodic sites). The metal ions pass into the solution and release electrons that travel to cathodic electropositive places. Thus, the corrosion properties of these organic coatings depend mainly on the galvanic current flow between the zinc particles and the steel substrate. For this reason, the zinc content in such coatings must be very high, 80 to 93 wt.% [26–29]. However, metallic zinc protects the steel substrate electrochemically (cathodic protection) only in the first phase of its action. Zinc reacts with oxygen, water, and carbon dioxide present in the air, forming corrosion products ($ZnO$, $Zn(OH)_2$ or $ZnCO_3$). These corrosion products cover the zinc particles (thereby reducing the conductivity of the zinc) and perfectly seal all of the pores in the paint film, creating a very compact and perfectly adherent layer applying a barrier protection mechanism that is resistant to normal atmospheric influences [4,30–33].

Environmental regulations of the European Union are constantly tightening the requirements regarding the use of zinc compounds as anti-corrosion additives in paint materials due to the potential harmfulness of zinc and zinc complexes to the aquatic environment. For ecological and economic reasons, there have therefore been efforts to reduce the zinc content in paint materials with other pigments, with the help of which high anti-corrosion efficiency would be achieved [3,34,35]. One of the alternatives is the use of organic pigments. Organic pigments are substances with distinct colour shades, almost insoluble in water and organic solvents, and they are characterized by resistance to UV and visible radiation and thermal stability. For the most part, they are prepared by converting water-soluble organic dyes into insoluble organic pigments, which are created either by removing solubilizing groups from the organic dye molecule or by creating an insoluble salt by replacing soluble metal ions with less soluble ones. Commonly used organic pigments include, for example, phthalocyanine pigments, pigments based on azo compounds, zinc salt based on nitroisophthalic acid, or N-benzosulphonyl anthranilic acid [36,37]. Organic pigments permeate the dispersion medium in the form of crystalline particles. The properties and behaviours of organic pigments therefore depend not only on their molecular structure but especially on their crystalline structure. An important parameter for the production of new organic pigments is their solubility. During the preparation of the pigment, an effort is made to make it minimally soluble in the medium, as even partial solubility can

cause subsequent recrystallization and thus a change in its properties [38]. Metal complex pigments are generally azo compounds formed by a layered crystal structure, in which the individual layers are bound with hydrogen bonds or metal ions. A very wide range of both organic and inorganic compounds is included here. Metal-azo complexes are among the most important molecules that can ensure the thermal stability of organic dyes. For this reason, they have received much attention in academic and applied research [39,40].

## 2. Materials

Anthranilic acid (pure), sodium chloride (pure), and ammonium sulphate were obtained from LACHEMA n.p., Brno, Czech Republic. Sodium hydroxide, magnesium chloride, potassium hydroxide, and sodium nitrite were obtained from Lach-Ner, s.r.o., Neratovice, Czech Republic. Hydrogen chloride, hydrochloric acid, nitric acid (pure), magnesium oxide (pure), xylene, and chloroform were obtained from PENTA s.r.o., Prague, Czech Republic. Calcium magnesium orthophosphate hydrate (industrial sample referred by the manufacturer as a zinc-free corrosion inhibitor) was obtained from Heubach GmbH, Langelsheim, Germany. 3-methyl-1-phenyl-1H-pyrazol-5-ol, N-(3-ethoxyphenyl)-3-hydroxy-2-naphthamide (naftol AS-PH), and 2-amino-5-nitrofenol (83.1%) were obtained from Synthesia a.s., Pardubice, Czech Republic. 2-naphthol (99%), urotropine, and 1,8-naphthalene anhydride were obtained from Sigma Aldrich, St. Louis, MO, USA. 1,4,5,8-naphthalene tetracarboxylic dianhydride was obtained from Fluorochem Limited, Hadfield, UK. Epoxy ester resin (WorléeDur D46) was obtained from Worlée-Chemie G.m.b.H., Lauenburg, Germany, and a sicative (Valirex Mix 835 D60) was obtained from 3P-Chem, s.r.o, Popovice, Czech Republic. Zinc was obtained from Radka International s.r.o., Lázně Bohdaneč, Czech Republic. Loctite EA 9466 (adhesive for pull-off tests) was obtained from Ulbrich Hydroautomtic s.r.o., Brno, Czech Republic.

## 3. Experimental Part

### 3.1. The Synthesis of Azo Dyes and Magnesium Complexes

The structures of azo dyes as ligands for complexation with $Mg^{2+}$ (so-called complexes forming monoazo dyes) are shown in Figure 1. The preparations, meaning diazotization and coupling reactions, have been described, for example, in the literature [40–44].

**Figure 1.** *Cont.*

**Figure 1.** Chemical structure of Dye I, Dye II, Dye III, and Dye IV.

3.1.1. Synthesis of Magnesium Complex Mg-Dye-I ($C_{34}H_{26}MgN_8O_6$)

Diazotization of Anthranilic Acid

　　Anthranilic acid (6.85 g; 50 mmol) was mixed with 50 $cm^3$ of water. Then, 35% HCl (10.4 g; 100 mmol) was added. The mixture was externally cooled to 0–5 °C, and aqueous 2M $NaNO_2$ (27 $cm^{-3}$) was slowly added dropwise with stirring at 0–5 °C. The formed solution of diazonium compound I (Figure 2) was immediately used for a coupling reaction.

**Figure 2.** Summary equation of diazotization of anthranilic acid.

Coupling to 5-Methyl-2-phenyl-3-pyrazolone, Dye I ($C_{17}H_{14}N_4O_3$)

　　5-methyl-2-phenyl-3-pyrazolone (8.71 g, 50 mmol) was mixed with 80 $cm^3$ of water and 4 M NaOH (12.5 $cm^3$) was added under the stirring. Then, the formed solution was cooled to 0–5 °C, and the diazonium compound I was slowly added under vigorous stirring. The reaction pH was adjusted to a value of 8 (using 4 M NaOH). When the coupling reaction (Figure 3) was completed, the pH was adjusted using 30% HCl to the value of 1–2 (–COONa– COOH). Finally, the formed yellow product (Dye I) was filtered off and washed with water to remove salts and acids. Finally, Dye I was dried at 80 °C.

**Figure 3.** Summary equation of coupling reaction to Dye I.

Preparation of Magnesium Complex Mg-Dye-I ($C_{34}H_{26}MgN_8O_6$)

Dye I (14.45 g, 45 mmol) was mixed with water (200 cm$^3$), and MgCl$_2$ (4.285 g, 50 mmol) was added. The reaction mixture was then vigorously stirred at 80 °C for 4 h, and during the reaction, the pH was adjusted to 6 with NaOH solution. When the reaction (Figure 4) was completed, the reaction mixture was cooled at room temperature (22 °C), and the formed Mg-Dye-I was filtered off and washed with water to remove all salts (NaCl, etc.). The yield of Mg-Dye I was 16.69 g (68.3% relative to default Dye I).

**Figure 4.** Synthesis of Mg-Dye-I (for simplicity, metal complex bonds are not indicated in the structural formula).

### 3.1.2. Synthesis of Magnesium Complex Mg-Dye-II ($C_{26}H_{19}MgN_3O_5$)

Coupling Reaction to Naphthol AS-PH, Dye II ($C_{26}H_{21}N_3O_5$)

Naphthol AS-PH (15.3 g; 50 mmol) was mixed with water (150 cm$^3$), and 4 M NaOH (13.5 cm$^3$) was added. The mixture was heated at 60–80 °C under stirring to obtain a solution. Then, the reaction mixture was externally cooled to 0–5 °C, and CH$_3$OH (30 cm$^3$) was added under vigorous stirring. Then, the solution of the diazonium compound I was slowly added, and the reaction pH during the coupling reaction was maintained at a value of 8–9 (with 4 M NaOH), while the reaction temperature was 0–5 °C. The resulting red product Dye-II (Figure 5) was filtered off and dried at 80 °C. The yield of Dye II was 19.72 g (86.7% relative to default naphthol AS-PH).

Preparation of Magnesium Complex Mg-Dye-II ($C_{26}H_{19}MgN_3O_5$)

Dye-II (22.75 g, 50 mmol) was mixed with water (200 cm$^3$), and MgCl$_2$ (4.285 g, 50 mmol) was added. The reaction mixture was then vigorously stirred at 80 °C for 4 h, and during the reaction, the pH was adjusted to 6 with NaOH solution. When the reaction was completed (Figure 6), the reaction mixture was cooled at room temperature (22 °C), and the formed Mg-Dye-II was filtered off and washed with water to remove all salts (NaCl, etc.). The yield of complex Mg-Dye-II was 20.42 g (85.6% relative to default Dye II).

**Figure 5.** Summary equation of coupling reaction to Dye II.

**Figure 6.** Synthesis of Mg-Dye-II.

### 3.1.3. Synthesis of Magnesium Complex Mg-Dye-III ($C_{17}H_{10}MgN_2O_3$)

Coupling Reaction to 2-Naphthol, Dye III ($C_{17}H_{12}N_2O_3$)

2-naphthol (7.2 g; 50 mmol) was mixed with 100 cm$^3$ of water, and 4 M NaOH (12.5 cm$^3$) was added. The mixture was heated under stirring at 60–80 °C to obtain a solution. Then, the reaction mixture was externally cooled to 0–5 °C under vigorous stirring. Subsequently, the solution of the diazonium compound I was slowly added, the reaction pH was during the coupling reaction maintained at a value of 8–9 (with 4M NaOH),

and the reaction temperature was 0–5 °C. When the coupling was completed (Figure 7), the pH was adjusted to 2–3 using 35% HCl, and the solution was stirred for another hour. The resulting dark red product Dye-III was then filtered off and dried at 80 °C. The yield of Dye III was 13.13 g (89.9% relative to default 2-naphthol).

**Figure 7.** Summary equation of coupling reaction to Dye III.

Preparation of Magnesium Complex Mg-Dye-III ($C_{17}H_{10}MgN_2O_3$)

Dye III (12.86 g, 44 mmol) was mixed with water (200 cm$^3$) and MgCl$_2$ (4.189 g, 50 mmol), the reaction mixture was then stirred vigorously and heated to 80 °C for 4 h, and the pH was adjusted to 6 with NaOH solution. The reaction mixture was cooled at room temperature, and the resulting dark red-coloured complex Mg-Dye III (Figure 8) was filtered off and washed with water to remove all salts (NaCl, etc.). The yield of complex Mg-Dye-III was 13.50 g (97.5% relative to default Dye III).

**Figure 8.** Synthesis of Mg-Dye-III.

### 3.1.4. Synthesis of Magnesium Complex Mg-Dye-IV ($C_{25}H_{18}MgN_4O_6$)

Diazotization of 2-Amino-5-nitrophenol

2-amino-5-nitrophenol (83.1%, 9.28 g; 50 mmol) was mixed with water (100 cm$^3$), 35% HCl (13 g), and CH$_3$COOH (100%, 10 g). The mixture was heated under stirring at 70 °C to obtain very fine dispersion. Then, the mixture was externally cooled to 0–5 °C, and aqueous 2 M NaNO$_2$ (27 cm$^3$) was slowly added dropwise under vigorous stirring. During the reaction, CH$_3$OH (5 cm$^3$) was added. The formed dispersion of diazonium compound II (Figure 9) was immediately used for a coupling reaction.

**Figure 9.** Summary equation of diazotization of 2-amino-5-nitrophenol.

Coupling to Naphthol AS-PH, Dye-IV ($C_{25}H_{20}N_4O_6$)

Naphthol AS-PH (15.3 g; 50 mmol) was mixed with water (150 cm$^3$) and 4M NaOH (13.5 cm$^3$). The mixture was heated under vigorous stirring at 60–80 °C. Then, the mixture was externally cooled at 0–5 °C, and CH$_3$OH (30 cm$^3$) was added. Consequently, a dispersion of diazonium compound II was slowly added. During the coupling reaction, the reaction pH was maintained at a value of 8–9 (with 4 M NaOH solution). When the coupling was completed (Figure 10), the resulting dark blue product Dye-IV was filtered off and dried at 80 °C. The yield of Dye IV was 16.11 g (68.3% relative to default Naphthol AS-PH).

**Figure 10.** Summary equation of coupling reaction to Dye IV.

Preparation of Magnesium Complex Mg-Dye-IV ($C_{25}H_{18}MgN_4O_6$)

Dye-IV (21 g, 44 mmol) was mixed with water (200 cm$^3$), and MgCl$_2$ (4.189 g, 50 mmol) was added. Then, the reaction mixture was vigorously stirred and heated to 80 °C for 4 h, and the reaction pH was maintained at the value of 6 with 4 M NaOH solution. When the reaction was completed (Figure 11), the reaction mixture was cooled at room temperature (22 °C), and the formed dark blue complex Mg-Dye-IV was filtered off and washed with water to remove all salts (NaCl, etc.). The yield of complex Mg-Dye-IV was 21.50 g (99.2% relative to default Dye IV).

**Figure 11.** Synthesis of Mg-Dye-IV.

*3.2. Characterization of the Synthesized Organic Dyes and Pigments (Complexes) by Analytical Methods*

3.2.1. Inductively Coupled Plasma Optical Emission Spectroscopy (ICP-OES)

The determination of magnesium was performed using samples solubilized in a microwave digestion system (SpeedwaveXpert, Berghof, Tübingen, Germany). To a precisely weighted 0.05 to 0.2 g of sample, 7 mL of nitric acid was added, the mixture was left for 20 min in an open vessel for digestion, and then decomposition was conducted (170 °C for 15 min, 200 °C 20 min), after which the mineralized sample was transferred into a 50-mL final volume.

The determination of magnesium in the mineralized sample was performed using an inductively coupled plasma optical emission spectrometer (GBC, Dandenong, Victoria, Australia) equipped with a concentric nebulizer and cyclonic spray chamber (Glass Expansion, Melbourne, Australia) at the spectral line of Mg II 280.270 nm. The working conditions for ICP-MS analysis were the following: sample flow rate of 1.5 mL min$^{-1}$, plasma power input of 1000 W plasma, external and sample gases at 10, 0.6, and 0.65 L·min$^{-1}$, the voltage on the photomultipliers at 600, observation height of 6.5 mm, and three repeated measurements with a 1-s reading signal fixed background correction. Calibration standards of 10–5–1–0.5 to 0.1 mg·dm$^3$ were prepared from commercially available standard solutions of magnesium 1 g·dm$^{-3}$ (SCP, Science, Clark Graham, Baie D'Urfé, QC, Canada). Limits

of detection (concentration equivalent to three times the standard deviation of noise in place of the background correction) for both elements were around 0.1 $\mu g \cdot dm^{-3}$, while respecting preparation of the sample for analysis (sample weight 0.05 g and 50 mL), the limit of detection for the whole analytical procedure was 2 $mg \cdot kg^{-1}$.

### 3.2.2. Elemental Analysis

Elemental analyses of synthesized compounds azo dyes and synthesized compounds magnesium complex were performed on a Flash 2000 CHNS Elemental Analyzer (Thermo Fisher Scientific, Milan, Italy).

### 3.2.3. Energy Dispersive X-ray Spectroscopy (EDX)

Measurements are reported in wt.%. Scanning electron microscopy (SEM, TESCAN, VEGA 3, EasyProbe, Brno, Czech Republic) in conjunction with an energy-dispersive x-ray spectroscopic analyser (EDX) was used to examine the surface and chemical compositions. The typical measurement error for EDX is 1 at.%. The average of the three EDX measurements per sample was often used.

### 3.2.4. Mass Spectrometry—MALDI

High-resolution MALDI mass spectrometry was performed using the LTQ Orbitrap XL instrument (Thermo Fisher Scientific, Bremen, Germany). The "dried droplet" method was utilized for sample preparation. The instrument, equipped with a nitrogen UV laser operating at 337 nm and 60 Hz, was operated in positive-ion mode with a mass range of m/z 100–2000. The resolution was set at 100,000 at m/z 400.

To ensure consistent laser shot positions, predefined spiral plate motion patterns were employed. The matrix used for sample preparation consisted of a 0.2 M solution of 2,5-dihydroxybenzoic acid (DHB) in a mixture of 95% acetonitrile and 5% water. The matrix-to-sample molar ratio was approximately 40:1.

For data acquisition, the mass spectra were averaged over the entire recording to improve the signal-to-noise ratio.

### 3.2.5. X-ray Diffraction

Powder diffractograms were measured using a D8 ADVANCE X-ray diffractometer (Bruker AXS, Karlsruhe, Germany) fitted with a vertical $\Theta$-$\Theta$ goniometer (radius = 217.5 mm). The goniometer has a scintillation Na(Tl)I detector, a graphite secondary monochromator, and an X-ray tube with a Cu anode (U = 40 kV, I = 30 mA; = 1.5418). The measurements were performed at room temperature in the 2–50° (2$\Theta$) range with a 0.02° step and a reading duration of 5 s/step for the diffracted radiation intensity.

### 3.2.6. Infrared Spectroscopy (IR)

Infrared spectra were recorded in a 4000- to 400-$cm^{-1}$ region on a Bruker VERTEX 70v (with Platinum-ATR-unit A225) FTIR spectrometer using a diamond ATR crystal (resolution 4 $cm^{-1}$).

### *3.3. Characterization of the Studied Pigments and the Used Binder by Methods Used in the Field of Paint Materials*

### 3.3.1. Pigment Parameter Determination

The density of four types of synthesized organic pigments and three types of inorganic pigments was determined using a Micromeritics AutoPycnometer 1340 (Norcross, GA, USA). Oil absorption of studied pigments was measured by the "pestle mortar". The measured results of the above determinations were used to calculate the critical pigment volume concentration (CPVC).

### 3.3.2. SEM and EDX Measurements of Pigments

The scanning electron microscopy (SEM) scans and elemental composition data (EDX) of studied pigments were obtained using a LYRA 3 (Tescan, Brno, Czech Republic) scanning electron microscope equipped with an Aztec X-Max 20 EDS analyser (Oxford Instruments, Oxford, UK). Samples were coated with a 20-nm carbon conductive layer (ACE 200, Leica, Wetzlar, Germany) and measured on five $300 \times 300$ μm areas at 20 kV of accelerating voltage. The results were averaged, and the error bars represent standard deviations of measured values. The pigments were coated with 18 nm of a gold conductive layer as well, and SEM scans of the studied samples were acquired at 10 kV of acceleration voltage.

### 3.3.3. Specification of the Binder for Coatings

An epoxy ester resin with the commercial name WorléeDur D 46 (Worlée-Chemie G.m.b.H., Lauenburg, Germany) was used as a binder and for the preparation of model paint. The binder specification was as follows: oil—40%, EP-resin—60%, colour DIN ISO 4630, Gardner—max. 10, acid value DIN EN ISO 2114: max. 4 [mgKOH/g], viscosity, rheometer, 23 °C, C 60/2, 50 s: 3.6–4.8 Pa·s$^{-1}$.

### 3.4. Formulation and Preparation of the Organic Coatings

Four types of synthesized organic pigments ($C_{34}H_{26}MgN_8O_6$, $C_{26}H_{19}MgN_3O_5$, $C_{17}H_{10}MgN_2O_3$, and $C_{25}H_{18}MgN_4O_6$) and two types of inorganic pigments (MgO and CaMg-HPO$_4$) were used together with zinc to formulate model paint. The organic coatings were formulated at PVC$_{pigment}$ = 1, 3, 5 and 10%. The model paints were also pigmented with metal zinc with a spherical particle shape to maintain a constant concentration of solids in which PVC/CPVC = 0.60. The model paints were prepared using a Dissolver type system at 3000 rpm/45 min.

After preparation, the prepared model paints were applied to standard steel panels (type S-46, S-36 and QD-24 low-carbon steel panels, Q-Lab Corporation (Cleveland, OH, USA), ISO Panel Specifications: 1514 Section 3.5.4). The dry film thickness (DFT) of the prepared coating films was measured by magnetic gauge (byko-test 8500 premium Fe/NFe, BYK Additives & Instruments, Wesel, Germany), according to ISO 2808. The vertical cut in the organic coating (studied in accelerated cyclical corrosion tests) was 100 mm long and 1 mm wide and was made in accordance with CSN EN ISO 12944-6 using a cutting tool (Elcometer 1538, DIN scratching tool with 1-mm cutter, Manchester, UK), complying with ISO 2409.

### 3.5. Mechanical Properties of the Paints

The mechanical properties of the studied organic coatings were studied based on the procedures specified in the following standards: ISO 2409 Cross-cut test, ISO 6272 Rapid-deformation (impact resistance) tests, ISO 1520 Cupping test, ISO 1519 Bend test (cylindrical mandrel), and ISO 4624 Pull-off test for adhesion [45,46].

### 3.6. Corrosion Test Procedures and Evaluation of Results after Corrosion Tests

### 3.6.1. Accelerated Cyclic Corrosion Testing in an Atmosphere of NaCl with Water Steam Condensation (Derived from ISO 9227)

In a testing chamber (SKB 400 A-TR-TOUCH, Gebr. Liebisch GmbH & Co. KG, Bielefeld, Germany), testing was performed in 12-h cycles divided into three parts: 10 h of exposure to a mist of 5% solution of NaCl at a temperature of 35 °C; 1 h of exposure at a temperature of 23 °C; and 1 h of humidity condensation at a temperature of 40 °C. The samples were exposed to these conditions and evaluated after 1440 h of exposure.

### 3.6.2. Accelerated Cyclic Corrosion Testing in an Atmosphere of SO$_2$ with Water Condensation (ISO 6988)

Samples were exposed to a 24-h cycle in a closed chamber (KB 300A, Gebr. Liebisch GmbH & Co. KG, Bielefeld, Germany): 8 h of exposure to SO$_2$ at a temperature of 38 °C

(1000 mL of sulphur dioxide was injected into a 300-L chamber), followed by exposure to the condensation of humidity for 16 h at a temperature of 21 °C. The condition of the samples was evaluated after 960 h of exposure.

### 3.6.3. Accelerated Cyclic Corrosion/Weather Resistance Testing with Exposure to a Salt Electrolyte and UV Radiation

A 12-h cycle of testing was performed, divided into three parts (SKB 400 A-TR-TOUCH, Gebr. Liebisch GmbH & Co. KG, Bielefeld, Germany): 10 h of exposure to a mist of 0.05%-solution of NaCl + 0.35%-solution of $(NH_4)_2SO_4$ at a temperature of 35 °C; 1 h of exposure at a temperature of 23 °C; and 1 h of humidity condensation at a temperature of 40 °C. Evaluation of samples was carried out after 168 hrs. The samples were then exposed in a fluorescent UV/condensation chamber (QUV/se Tester, Q-Panel Lab Products, Cleveland, OH, USA) in 12-h cycles divided into two parts: a cycle consisting of 8 h of exposure to UV radiation at 60 °C (using UVA-340 discharge lamps), followed by 4 h of exposure to moisture at 50 °C. The samples were evaluated after 168 h of exposure. This order of testing was repeated three times, making the total exposure time 1008 h. The procedure is derived from ASTM D 5894-96.

### 3.6.4. Evaluation of Samples after Corrosion Tests

The following standards describe the evaluation of corrosion in samples after corrosion tests: ASTM D714-87 Standard Test Method for Evaluating Degree Of Blistering Of Paints; and ASTM D 610-85 Method for Evaluating Degree of Rusting on Painted Steel Surfaces [47,48].

### *3.7. Electron Microanalysis Studied Organic Coatings*

After 1440 h of exposure to the sale atmosphere, the studied organic coatings were examined by electron microanalysis to ascertain the elemental composition of the organic coatings containing the studied organic and inorganic pigments using a TESCAN VEGA 5130SB scanning electron microscope and a Bruker Quantax 200 energy dispersive X-ray spectrometer.

### *3.8. Electrochemical Methods*

A cell having three-electrode geometry, in which a paint-coated sample acts as a test electrode (6 $cm^2$), platinum as a counter electrode, and a saturated calomel electrode as a reference electrode, was used. The cell was connected to a Gamry Interface 1000 (Wilmington, DE, USA) for the measurements in 3.5 mass% NaCl solution. Tafel curves were obtained by applying potential from $E_{corr}$ to 250 mV in cathodic and anodic directions. Gamry Echem Analyst $^{TM}$ software (version, year) was used to determine the corrosion rate. Long-term performance of the painted samples was assessed by impedance spectroscopy over a frequency range of 100 kHz to 0.1 Hz using an amplitude signal of 10 mV rms per ASTM G 106 and ASTM 2005b. All of the measurements were carried out five times to obtain good repeatabilty of the results.

### *3.9. Determination of pH and Specific Electrical Conductivity and Corrosion Loss from Aqueous Extracts of Pigments and of Loose Paint Films*

For the determination of pH and specific electrical conductivity ($\lambda$) from aqueous extracts of pigments, 10% suspensions of pigments were prepared in redistilled water, and a loose paint film was prepared [49,50]. The loose paint films were prepared by applying paint to polyethylene foils, from which they were removed after 60 days. The loose paint films were cut to a size of 1 mm × 1 mm. The pH and specific electrical conductivity of the prepared suspensions were measured over the course of 28 days. pH was measured using a pH meter (WTW 320, Labexchange-Die Laborgerätebörse GmbH, Burladingen, Germany) according to the procedure derived from CSN EN ISO 787-9. The pH meter was calibrated before the determination against buffer solutions of known pH values at the temperature of the test. Specific electrical conductivity was measured using a conductometer (Handylab

LF1, Schott-Geräte GmbH, Landshut, Germany) according to the procedure derived from CSN EN ISO 787-14. The conductometer was calibrated before the determination against standard calibration solutions of known electrical conductivity values at the temperature of the test. After 28 days, both types of suspensions were filtered, and the filtrates were used to determine the corrosion losses when steel panels (40 mm × 20 mm × 1 mm) were cleaned, weighed on an analytical balance with an accuracy of ±0.0001 g, and immersed in aqueous extracts. The determination of corrosion loss was completed after 7 days when the steel panels were removed from the filtrates, freed from corrosion products (aqueous solution of 20% HCl with an addition of 0.5 wt.% urotropine), dried, and weighed again. The mass values were used to calculate respective losses caused by corrosion ($K_m$). The losses were expressed in weight percentages ($X_H$) related to the weight loss of steel in distilled water. Based on these results, loss of dimensions of steel panels ($U_R$) and weight losses of steel ($V_K$) were calculated [51,52].

## 4. Results and Discussion

### 4.1. Synthesis and Characterization of New Azo Organic Pigments by Analytical Methods

Four azo dyes (ligands) were prepared by diazotization and coupling reactions, followed by complexation with magnesium cation. All diazonium compounds were prepared by so-called direct diazotization using aqueous sodium nitrite in dilute hydrochloric acid. Coupling reactions were carried out by the addition of the diazonium salt previously prepared as a solution of the secondary component at 0–5 °C and a reaction pH of 8–9. Afterwards, the resulting ligand and magnesium chloride were mixed in water and heated to 80 °C under vigorous stirring. The precipitated solid was collected by filtration, washed with water, and then dried to obtain magnesium complexes. After separation, the final yield of products ranged from 68.3% to 99.2%. The synthetic procedures are described in experimental Section 3.1.

Thus, the four azo dyes were synthesized by a diazo-coupling reaction to form Dye I, Dye II, Dye III, and Dye IV, the reaction yields of which are shown in Table 1. Consequently, the four magnesium complexes Mg-Dye-I, Mg-Dye-II, Mg-Dye-III, and Mg-Dye-IV were prepared by the reaction of Dye-I, Dye-II, Dye-III, and Dye-IV with magnesium chloride, and the reaction yields are shown in Table 2. The prepared new anticorrosive "magnesium-azo-pigments" were characterized by analytical methods. The energy-dispersive X-ray spectroscopy (EDX) analysis shows the expected magnesium signals and confirmed the existence of magnesium elements in the azo-carboxylate ligand, while the distribution of elements is clarified by the EDX results, which show that magnesium elements are homogeneously distributed in the whole sample. The determination of magnesium (ICP-OES) analysis showed the expected magnesium signals and confirmed the content percentage of the magnesium composition of the azo-carboxylate ligand. The A D8 ADVANCE X-ray diffractometer (XRD) patterns of azo-carboxylate ligand showed strong diffraction peaks. These peaks were used to determine the approximate crystalline size using Scherrer's method.

**Table 1.** Characterization of synthesized azo dye compounds.

| Compound | Chemical Formula | Molecular Weight [g·mol⁻¹] | Elemental Analysis (Found/Calculated) [%] | Yield [%] |
|---|---|---|---|---|
| Dye-I | $C_{17}H_{14}N_4O_3$ | 322.32 | C—58.21/63.35<br>H—3.84/4.38<br>N—15.69/17.38 | 97.4 |
| Dye-II | $C_{26}H_{21}N_3O_5$ | 455.47 | C—62.53/68.56<br>H—4.11/4.65<br>N—8.20/9.23 | 86.7 |
| Dye-III | $C_{17}H_{12}N_2O_3$ | 292.29 | C—70.48/69.86<br>H—3.99/4.14<br>N—8.14/9.58 | 89.9 |
| Dye-IV | $C_{25}H_{20}N_4O_6$ | 472.46 | C—60.92/63.56<br>H—4.39/4.27<br>N—11.08/11.86 | 68.3 |

**Table 2.** Characterization of synthesized compounds of magnesium complex.

| Compound | Chemical Formula | Molecular Weight [g·mol$^{-1}$] | Elemental Analysis (Found/Calculated) [%] | Yield [%] | Content of Mg (Prepared/Theoretical) [mg·kg$^{-1}$] |
|---|---|---|---|---|---|
| Mg-Dye-I | $C_{34}H_{26}MgN_8O_6$ | 666.90 | C—56.70/58.09<br>H—4.36/4.30<br>N—15.28/15.94<br>Mg—3.46/4.14 | 68.3 | 34,600/41,400 |
| Mg-Dye-II | $C_{26}H_{19}MgN_3O_5$ | 477.76 | C—62.61/65.36<br>H—4.63/4.01<br>N—8.17/8.80<br>Mg—5.59/5.09 | 85.6 | 55,900/50,900 |
| Mg-Dye-III | $C_{17}H_{10}MgN_2O_3$ | 314.58 | C—54.06/58.24<br>H—3.77/4.02<br>N—7.20/7.99<br>Mg—7.15/6.93 | 97.5 | 71,500/69,300 |
| Mg-Dye-IV | $C_{25}H_{18}MgN_4O_6$ | 494.75 | C—53.67/56.57<br>H—4.12/4.18<br>N—9.67/10.56<br>Mg—4.32/4.58 | 99.2 | 43,200/45,800 |

### 4.2. Maldi Characterization

The MALDI-HRMS results are presented below. Because the spectra were measured in the positive mode, the positive ion of the dye is always present as one of the dominant peaks. The matrix typically provides positive ions as adducts with sodium cations, protons, or all of the matrix. The presence of the adducts listed in the text is common. Degrading such complexes and replacing the metal in the adduct are always performed. We also found some peaks with small intensity, corresponding to positive ions containing magnesium.

*Dye-I*

HR-MALDI-MS (DHB): calcd for $C_{17}H_{14}N_4O_3$ m/z 323.1139 ([M+H]), found 323.1144.
HR-MALDI-MS (DHB): calcd for $C_{17}H_{14}N_4O_3$ m/z 345.0958 ([M+Na]), found 345.0963.
HR-MALDI-MS (DHB): calcd for $C_{17}H_{14}N_4O_3$ m/z 367.0778 ([M+2Na]), found 367.0783.

*Dye-II*

HR-MALDI-MS (DHB): calcd for $C_{26}H_{21}N_3O_5$ m/z 456.1554 ([M+H]), found 456.1562.
HR-MALDI-MS (DHB): calcd for $C_{26}H_{21}N_3O_5$ m/z 478.1373 ([M+Na), found 478.1381.

*Dye-III*

HR-MALDI-MS (DHB): calcd for $C_{17}H_{12}N_2O_3$ m/z 293.0921 ([M+H]), found 293.0924.
HR-MALDI-MS (DHB): calcd for $C_{17}H_{12}N_2O_3$ m/z 315.0740 ([M+Na]), found 315.0744.

*Dye-IV*

HR-MALDI-MS (DHB): calcd for $C_{25}H_{20}N_4O_6$ m/z 473.1456 ([M+H]), found 473.1464.
HR-MALDI-MS (DHB): calcd for $C_{25}H_{20}N_4O_6$ m/z 495.1275 ([M+Na), found 495.1282.

*Mg-Dye-I*

HR-MALDI-MS (DHB): calcd for $C_{34}H_{26}MgN_8O_6$ m/z 667.1899 ([M+H]), found 667.1915.
HR-MALDI-MS (DHB): calcd for $C_{34}H_{26}MgN_8O_6$ m/z 689.1718 ([M+Na]), found 689.1735.

*Mg-Dye-II*

HR-MALDI-MS (DHB): calcd for $C_{26}H_{19}MgN_3O_5$ m/z 478.1248 ([M+H]), found 478.1263.
HR-MALDI-MS (DHB): calcd for $C_{26}H_{22}N_3O_5$ m/z 456.1554 ([M+H]), found 456.1566.
HR-MALDI-MS (DHB): calcd for $C_{26}H_{19}N_3O_5$ m/z 478.1373 ([M+Na]), found 478.1386.

*Mg-Dye-III*

HR-MALDI-MS (DHB): calcd for $C_{17}H_{12}N_2O_3$ m/z 315.0740 ([M+Na]), found 315.0744.

*Mg-Dye-IV*

HR-MALDI-MS (DHB): calcd for $C_{25}H_{20}N_4O_6$ m/z 495.1275 ([M+Na]), found 495.1287.

### 4.3. Energy Dispersive X-ray Spectroscopy (EDX)

The expected magnesium signals from the energy-dispersive X-ray spectroscopy analysis were confirmed by the magnesium elements' presence in the magnesium complexes. The distribution of the elements was also made clear by the EDX results, which demonstrated that the magnesium elements were uniformly distributed throughout the entire sample.

### 4.4. X-ray Powder Diffraction Patterns

X-ray powder diffraction patterns are shown in the following Figure 12. The stacking geometry of the crystalline phases of the magnesium complex was determined by side groups of different sizes. The pigments (Dye-I–Dye-IV and Mg-Dye-I–Mg-Dye-IV) showed their intense characteristic peaks located at 2θ values.

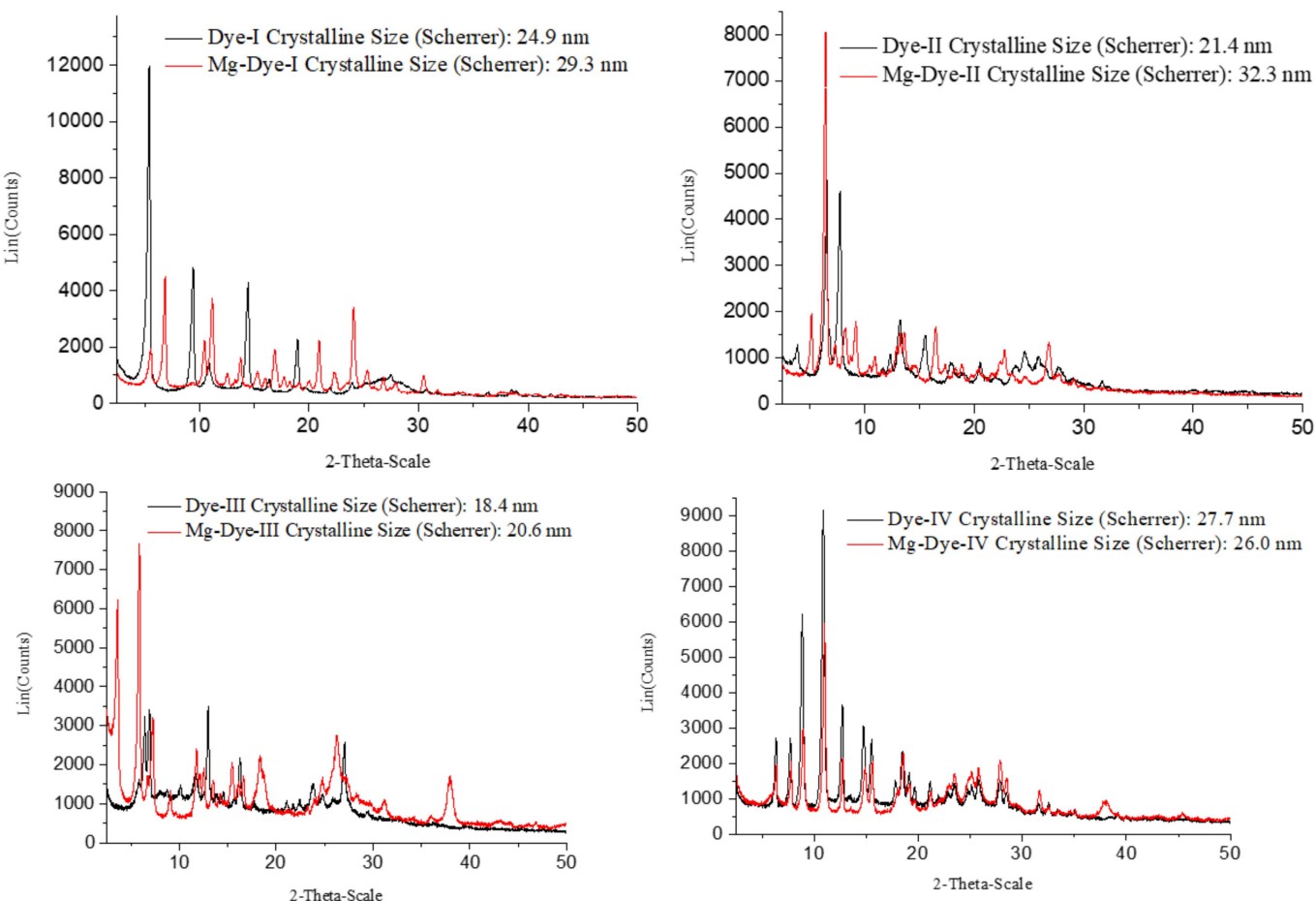

**Figure 12.** X-ray powder diffraction of the studied organic pigments.

Pigment Dye-I: 16.34, 9.40, 8.13, 6.14, 5.41, and 4.68°, with a Scherrer particle size of 24.9 nm.

Pigment Mg-Dye-I: 15.95, 12.90, 9.41, 8.45, 7.92, 6.43, 5.79, 5.25, 4.64, 4.24, and 3.69°, with a Scherrer particle size of 29.3 nm.

Pigment Dye-II: 23.03, 13.59, 11.42, 7.18, 6.70, 5.71, 4.96, 4.32, 3.73, 3.61, 3.43 and 3.22°, with a Scherrer particle size of 21.4 nm.

Pigment Mg-Dye-II: 17.24, 13.81, 12.10, 10.77, 9.63, 8.10, 7.21, 6.68, 6.50, 5.37, 3.90, and 3.32°, with a Scherrer particle size of 21.7 nm.

Pigment Dye-III: 15.14, 13.71, 12.77, 11.10, 8.70, 7.57, 6.81, 5.44, 4.21, 3.76, and 3.30°, with a Scherrer particle size of 18.4 nm.

Pigment Mg-Dye-III: 24.41, 15.05,12.16, 9.73, 7.49, 7.57, 7.06, 6.55, 5.73, 5.33, 4.83, 3.59, 3.39, and 2.37°, with a Scherrer particle size of 20.6 nm.

Pigment Dye-IV: 14.04, 11.50,10.01, 8.14, 6.98, 6.01, 5.72, 4.81, 4.64, 4.20, 3.79, 3.45, 3.20, and 3.13°, with a Scherrer particle size of 27.7 nm.

Pigment Mg-Dye-IV: 15.32, 13.95,11.41, 9.90, 8.06, 6.96, 5.96, 5.69, 4.78, 4.64, 4.19, 3.88, 3.45, 3.20, and 3.13°, with a Scherrer particle size of 26.0 nm.

### 4.5. Infrared Spectroscopy (IR)

The IR spectra of azo dyes (blue) and magnesium complexes (red) in transmittance mode are shown in Figure 13, where there are differences between free ligands and magnesium complexes.

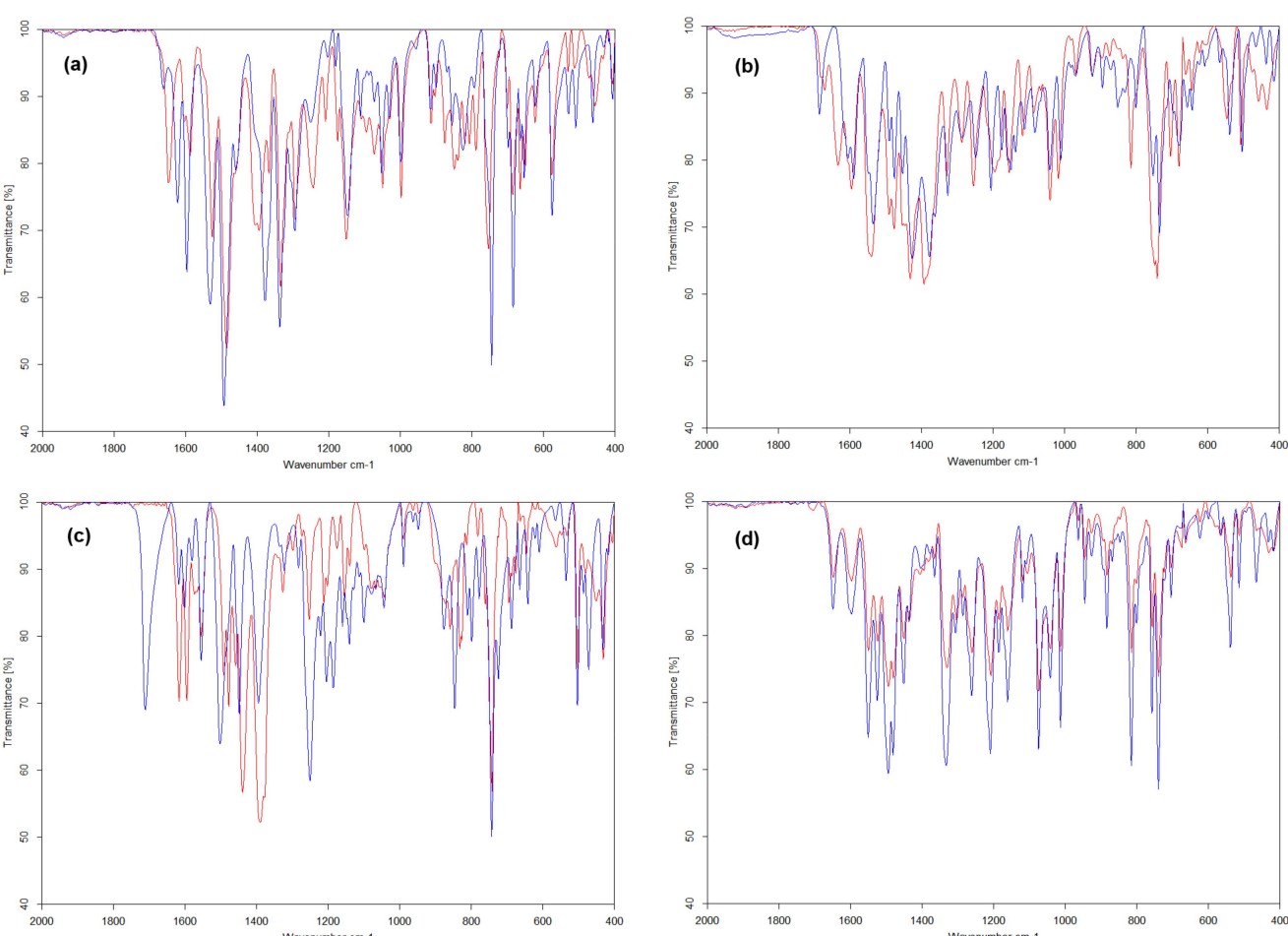

**Figure 13.** IR spectra of sets Dye-I–Mg-Dye-I (**a**), Dye-II–Mg-Dye-II (**b**), Dye-III–Mg-Dye-III (**c**), and Dye-IV–Mg-Dye-IV (**d**).

### 4.6. Characterization of the Studied Pigments

Pigment Specification

The studied pigments were subjected to measurements of the typical paint parameters, i.e., density and oil number, which are used to calculate of the CPVC parameters of each pigment. The results of the pigments studied are summarized in Table 3. The densities of the magnesium complexes lay within the range of 1.37 to 1.55 g·cm$^{-3}$; the oil numbers lay between 43.2 and 75.8 g/100 g of the pigment; and the CPVC values were 46–67. The magnesium oxide (MgO) density was 3.01 g·cm$^{-3}$; the oil number was 48.1 g/100 g of the pigment; and the CPVC was 36; while for CaMgHPO$_4$, the density was 2.72 g·cm$^{-3}$; the oil number was 37.5 g/100 g of the pigment; and the CPVC was 48. Due to the low density of

magnesium-containing pigments, they can be expected to have a markedly lower tendency to settle to the bottom of containers compared to zinc particles. The zinc density was 7.14 g·cm$^{-3}$; the oil number was 6.4 g/100 g of the pigment; and the CPVC was 67.

**Table 3.** Characteristics of the studied pigments: density, oil number, and critical pigment volume concentration (CPVC).

| Pigment | Density [g·cm$^{-3}$] | Oil Absorption [g/100 g] | CPVC [-] |
|---|---|---|---|
| Mg-Dye-I $C_{34}H_{26}MgN_8O_6$ | 1.38 ± 0.02 | 45.9 | 59 |
| Mg-Dye-II $C_{26}H_{19}MgN_3O_5$ | 1.37 ± 0.02 | 56.8 | 54 |
| Mg-Dye-III $C_{17}H_{10}MgN_2O_3$ | 1.55 ± 0.02 | 43.2 | 58 |
| Mg-Dye-IV $C_{25}H_{18}MgN_4O_6$ | 1.41 ± 0.02 | 75.8 | 46 |
| MgO | 3.01 ± 0.02 | 48.1 | 36 |
| Ca-Mg-HPO$_4$ | 2.72 ± 0.02 | 37.5 | 48 |
| Zn | 7.14 ± 0.02 | 6.4 | 67 |

Representative SEM scans of studied pigments are shown in Figure 14. The Mg-Dye-I sample consisted of needle- or sheet-like particles with a width of 100–500 nm and length of 0.5–2 μm. Their form was unique as the other prepared pigments possessed either spherical or undefined shapes. In the case of samples Mg-Dye-II, III and IV, amorphous shapes were prevalent with average particle diameters of 0.1–1 μm. The pigments also showed a higher tendency for agglomeration. MgO formed well-defined polyhedral nanoparticles 100 nm in diameter and with a narrow distribution of sizes. In contrast, the Ca-Mg-HPO$_4$ pigment possessed polyhedral particles with a quite wide distribution of sizes ranging from 100 nm to 1 μm. The Zn pigment formed large spherical particles 0.5–5 μm in diameter.

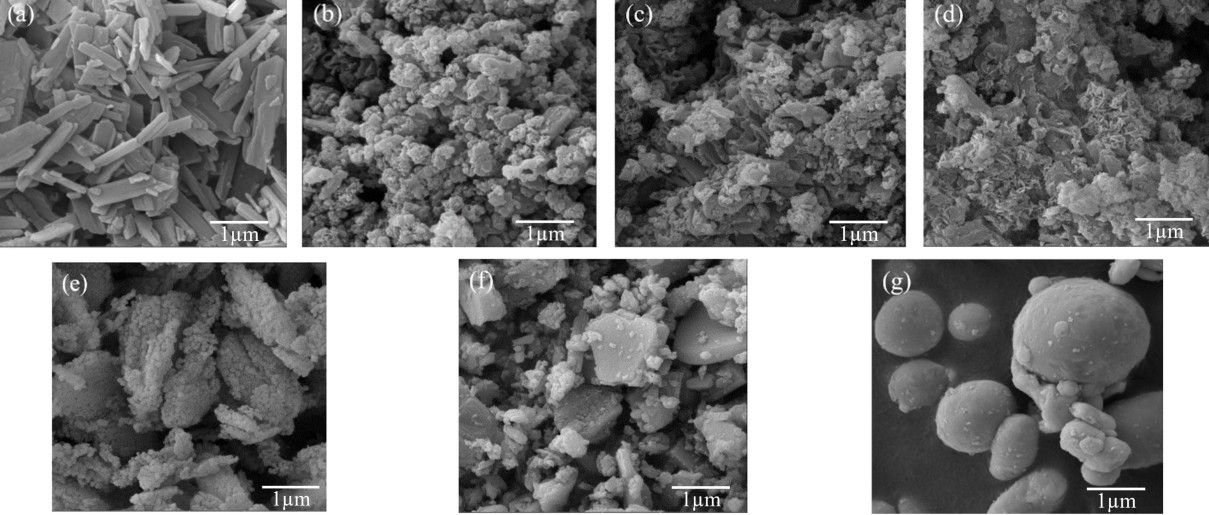

**Figure 14.** Scanning electron micrographs of the studied pigments: (**a**) Mg-Dye-I; (**b**) Mg-Dye-II; (**c**) Mg-Dye-III; (**d**) Mg-Dye-IV; (**e**) MgO; (**f**) Ca-Mg-HPO$_4$; (**g**) Zn.

The results of the EDX analysis of the studied pigments are summarized in Table 4. The five 300 × 300 μm spots were analysed for each sample, and the averaged values are presented in the table together with standard deviations. The large area of analysis and number of measurement spots ensured more representative data about the studied samples. The data confirmed that the synthesized organic pigments clearly contained significant numbers of Mg atoms. The carbon content was slightly artificially elevated due to the

necessary presence of a conductive carbon layer. The data also confirmed the occasional presence of Na ($C_{26}H_{19}MgN_3O_5$ and $C_{17}H_{10}MgN_2O_3$). The presence of these elements (especially Na) can be explained by the contamination of the synthesized complexes from the substances used for their synthesis. The data confirm that the inorganic pigment MgO contains significant numbers of Mg atoms and O atoms, while the inorganic pigment Ca-Mg-$HPO_4$ contains significant numbers of Ca, P, O, and Mg atoms.

**Table 4.** The results of the SEM-EDX analysis of the studied pigments obtained at five different spots. Error bars represents standard deviations of measured values.

| Complex | Element [Atomic %] | | | | | | |
|---|---|---|---|---|---|---|---|
| | C | N | O | Na | Mg | P | Ca |
| Mg-Dye-I $C_{34}H_{26}MgN_8O_6$ | 65.3 ± 0.9 | 15.1 ± 0.5 | 17.8 ± 0.7 | - | 1.8 ± 0.2 | - | - |
| Mg-Dye-II $C_{26}H_{19}MgN_3O_5$ | 68.9 ± 0.5 | 8.0 ± 0.4 | 20.2 ± 0.3 | 0.5 ± 0.1 | 2.4 ± 0.2 | - | - |
| Mg-Dye-III $C_{17}H_{10}MgN_2O_3$ | 68.2 ± 0.2 | 8.1 ± 0.3 | 20.4 ± 0.2 | 0.9 ± 0.2 | 2.4 ± 0.2 | - | - |
| Mg-Dye-IV $C_{25}H_{18}MgN_4O_6$ | 67.0 ± 1.0 | 8.6 ± 0.7 | 22.2 ± 0.5 | - | 2.2 ± 0.3 | - | - |
| MgO | - | - | 52.5 ± 0.7 | - | 47.5 ± 0.3 | - | - |
| Ca-Mg-HPO4 | - | - | 68.0 ± 0.5 | - | 1.5 ± 0.3 | 12.8 ± 0.5 | 17.7 ± 0.5 |

The XRD diffractograms of zinc, magnesium oxide, and Ca-Mg-$HPO_4$ are shown in Figure 15. This analysis (XRD) provided evidence that the zinc used to prepare the coatings was in the form of pure metal without the presence of other substances (zinc salts). The results of this analysed pigment MgO confirmed the dominant proportion of magnesium oxide, in addition to which the presence of magnesium hydroxide was also confirmed. The presence of other types of magnesium salts was not demonstrated in this diffractogram. These facts are also confirmed by the manufacturers of the given types of pigments on the technical data sheets. The diffractogram of the pigment marked with the abbreviation Ca-Mg-$HPO_4$ demonstrates the presence of the following compounds: $Ca_9HPO_4(PO_4)_5OH$ (calcium hydrogen phosphate hydroxide/calcium-deficient hydroxyapatite) [53], $Ca_5(PO_4)_3(OH)$ (hydroxylapatite), $CaPO_3(OH)$ (monetite), and $MgHPO_4 \cdot H_2O$ (newberyite). In the tables and text, this pigment is referred to below by the general formula Ca-Mg-$HPO_4$, as stated by the manufacturer. The presence of zinc compounds in this type of pigment was unequivocally excluded by this method, with this conclusion corresponding to the information given on the technical sheet of the relevant type of pigment.

*4.7. Results of Mechanical Properties of the Protective Coatings*

Five types of mechanical tests were used to assess and compare the mechanical properties of the individual coating films studied in accordance with the given types of standards. The adhesion (cross-cut) test, the pull off test, the bending test, the cupping test, and the impact strength all tested the coatings and achieved comparable resistance compared to a standard zinc-pigmented organic coating. In the adhesion (cross-cut) test, the lattice-patterned film surface remained intact for all the studied films. In the bending test, there was no damage observed in any films, even after using a mandrel with a diameter of 2 mm. The studied films also remained undamaged in the free drop test, in which a 1000-g weight was dropped from a 1-m height. In the cupping test, the studied coatings achieved the maximum possible resistance, and the film was not damaged even when the indentation value was higher than 100 mm. During the pull-off test, the coatings containing the studied pigments achieved higher resistance compared to the standard organic coating pigmented only with zinc. Coatings pigmented with four types of organic pigments and also two types of inorganic pigments achieved a pull-off strength >2 MPa, while a coating pigmented only with zinc achieved a pull-off strength of 1.81 MPa. Maximum values of pull-off strength of 2.97 MPa were achieved by coatings with MgO pigment content at OKP = 5 and 10%. The

fracture type noted in the evaluation of all coatings was cohesion in the coating. Hence, the pigmentation of the zinc coatings with these studied pigments (inorganic and organic types) was favourable as far as the mechanical properties are concerned.

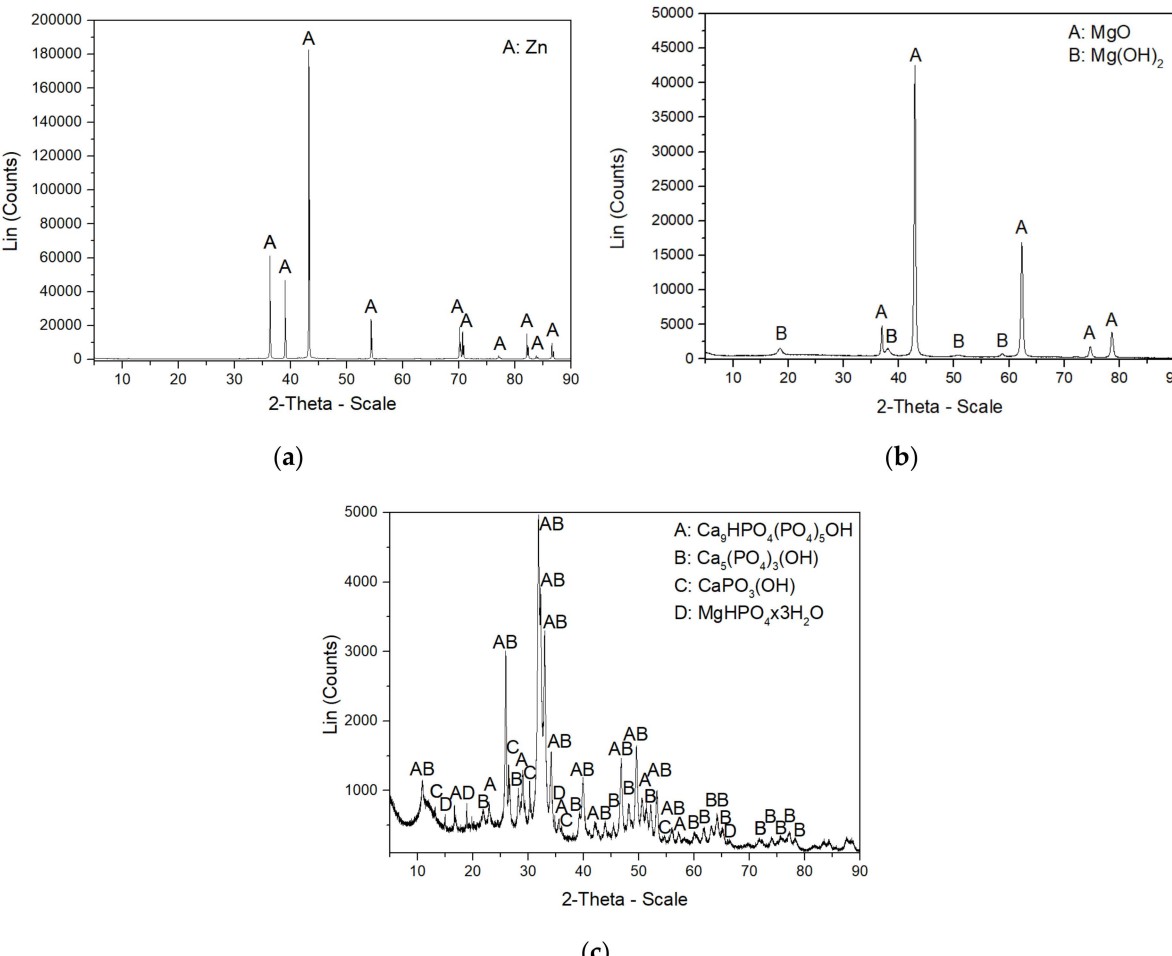

**Figure 15.** X-ray powder diffraction of the studied inorganic pigments (**a**) Zn, (**b**) MgO, and (**c**) Ca-Mg-HPO$_4$.

### 4.8. Results of Corrosion Tests of the Protective Coatings

4.8.1. Accelerated Corrosion Tests in a Salt Mist Atmosphere

On the steel panels with the investigated organic coatings, the corrosion manifestations due to the cyclic corrosion test in the neutral salt spray environment were evaluated. The organic coatings were subjected to this corrosion test for 1440 h, and the results are shown in Table 5. Photographs of the panels after 1440 h of exposure are presented in Figure 16.

**Table 5.** Results of corrosion testing performed in a salt mist chamber of the studied organic coatings containing studied pigments (PVC 1, 3, 5 and 10%) and zinc (PVC/CPVC = 0.60) after 1440 h of exposure, DFT = 90 ± 5 μm.

| Pigment | PVC [%] | Blistering | | Corrosion | |
| --- | --- | --- | --- | --- | --- |
| | | On the Film Area [dg] | In the Cut [dg] | In the Cut [mm] | Metal Base [%] |
| Mg-Dye-I $C_{34}H_{26}MgN_8O_6$ | 1 | 6F | 4M | 4–4.5 | 1 |
| | 3 | - | 6M | 4–4.5 | 0.1 |
| | 5 | - | 8M | 4–4.5 | 0.03 |
| | 10 | - | 8M | 3.5–4 | 0.03 |

**Table 5.** *Cont.*

| Pigment | PVC [%] | Blistering | | Corrosion | |
|---|---|---|---|---|---|
| | | On the Film Area [dg] | In the Cut [dg] | In the Cut [mm] | Metal Base [%] |
| Mg-Dye-II $C_{26}H_{19}MgN_3O_5$ | 1 | 8F | 2MD | 5.5–6 | 1 |
| | 3 | 8F | 2MD | 5.5–6 | 3 |
| | 5 | 8F | 2D | 6–6.5 | 3 |
| | 10 | 8F | 4M | 5–5.5 | 1 |
| Mg-Dye-III $C_{17}H_{10}MgN_2O_3$ | 1 | 8MD | 2MD | 6–6.5 | 100 |
| | 3 | 8D | 2D | 9–9.5 | 100 |
| | 5 | 8MD | 2M | 6–6.5 | 100 |
| | 10 | 8MD | 2M | 6.5–7 | 50 |
| Mg-Dye-IV $C_{25}H_{18}MgN_4O_6$ | 1 | 8M | 4M | 6–6.5 | 3 |
| | 3 | 8F | 2D | 6–6.5 | 10 |
| | 5 | 8M | 2D | 6–6.5 | 16 |
| | 10 | 8M | 4M | 6–6.5 | 16 |
| MgO | 1 | - | 4M | 6–6.5 | - |
| | 3 | 8F | 6M | 6–6.5 | 0.01 |
| | 5 | - | 6M | 5–5.5 | - |
| | 10 | - | 6M | 5–5.5 | - |
| CaMgHPO$_4$ | 1 | - | 4MD | 6.5–7 | 3 |
| | 3 | 8MD | 4MD | 6.5–7 | 50 |
| | 5 | 8MD | 4MD | 6.5–7 | 50 |
| | 10 | - | 2MD | 6–6.5 | 0.1 |
| Zn | - | - | 2D | 8–8.5 | 0.3 |

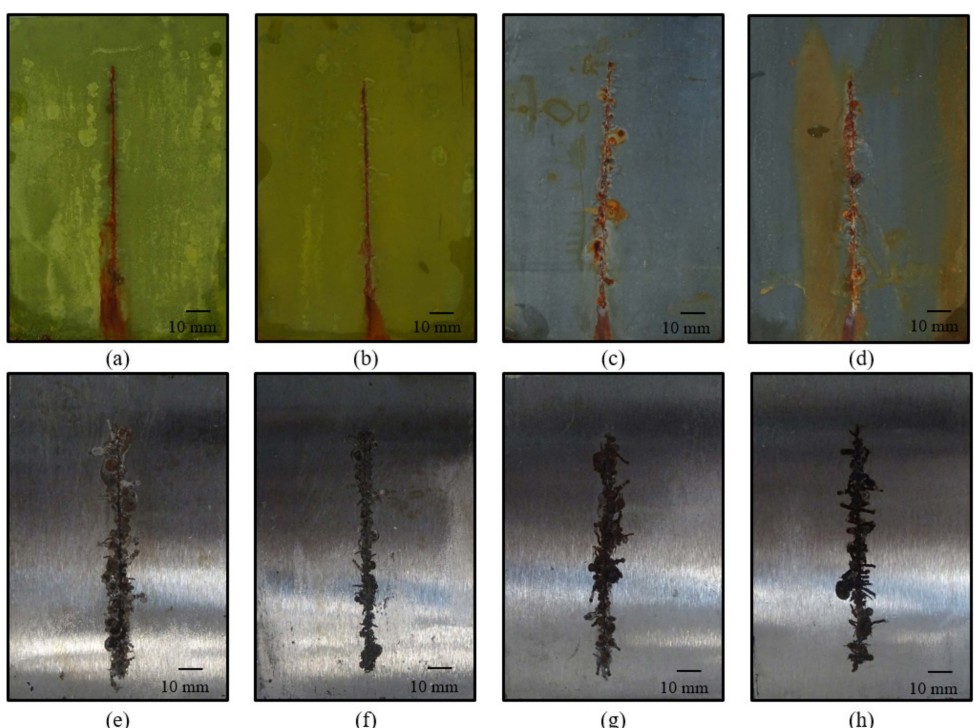

**Figure 16.** Organic coating after 1440 h of exposure in a salt mist atmosphere and steel panel before removing the organic coating. Organic coating with: (**a**) $C_{34}H_{26}MgN_8O_6$ at PVC = 5%, (**b**) $C_{34}H_{26}MgN_8O_6$ at PVC = 10%, (**c**) MgO at PVC = 5%, (**d**) MgO at PVC = 10%; steel panel after removing the organic coating with: (**e**) $C_{34}H_{26}MgN_8O_6$ at PVC = 5%, (**f**) $C_{34}H_{26}MgN_8O_6$ at PVC = 10%, (**g**) MgO at PVC = 5%, (**h**) MgO at PVC = 10%.

Among the organic coatings containing spherical zinc and organic pigments, the coatings with $C_{34}H_{26}MgN_8O_6$ pigment (Mg-Dye-I) achieved the highest efficiency, in which the resistance to corrosion propagation in the test cut increased, and the extent of corrosion in the coating film surface decreased with increasing PVC values; also, the number of blisters in the test cut decreased, and blisters in the coating film surface were found only at OKP = 1% (6F). The highest anti-corrosion efficiency was achieved by the coating with $C_{34}H_{26}MgN_8O_6$ pigment at a PVC = 10%, in which, after 1440 h of exposure, no blisters appeared on the surface of the paint film, corrosion around the test cut was in the range of 3.5–4 mm, the blisters in the test cut were graded 8M, and corrosion of the steel substrate was only 0.03%. In contrast, low anticorrosive efficiency was achieved by organic coatings with the organic pigment $C_{17}H_{10}MgN_2O_3$ (Mg-Dye-III), which at all PVC values showed a large number of blisters on the surface of the coating film (in the range of 8MD–8D); the corrosion resistance in the vicinity of the test cut was also low, and at PVCs = 1, 3 and 5%, the corrosion of the steel substrate was 100%. Among the organic coatings containing inorganic pigments, the highest corrosion efficiency after 1440 h of exposure was achieved by organic coatings with MgO pigment at PVCs = 5 and 10%, with no blisters in the coating film surface, while blisters in the test cut were rated 6M, and corrosion in the test cut was in the range of 5–5.5 mm. At the same time, no corrosion of the steel substrate occurred in these organic coatings.

4.8.2. Accelerated Corrosion Tests in Atmosphere Containing $SO_2$

The organic coatings were also subjected to a corrosion test in an atmosphere containing $SO_2$ for 1440 h, and the results are shown in Table 6. Photographs of the panels after 1440 h of exposure are presented in Figure 17.

**Table 6.** Results of corrosion testing performed in a condenser chamber filled with an atmosphere containing $SO_2$ of the studied organic coatings containing studied pigments (PVC 1, 3, 5 and 10%) and zinc (PVC/CPVC = 0.60) after 1440 h of exposure, DFT = 90 ± 10 µm.

| Pigment | PVC [%] | Blistering | | Corrosion | |
| --- | --- | --- | --- | --- | --- |
| | | On the Film Area [dg] | In the Cut [dg] | In the Cut [mm] | Metal Base [%] |
| Mg-Dye-I $C_{34}H_{26}MgN_8O_6$ | 1 | 6M | 6F | 0.5–1 | 0.3 |
| | 3 | 8M | 6F | 0.5–1 | 0.3 |
| | 5 | 8F | 8F | 0.5–1 | 0.1 |
| | 10 | 8F | 6F | 0–0.5 | 0.1 |
| Mg-Dye-II $C_{26}H_{19}MgN_3O_5$ | 1 | 8MD | 8F | 0.5–1 | 0.3 |
| | 3 | 6F | 6F | 0.5–1 | 0.3 |
| | 5 | 8MD | 8MD | 0–0.5 | 0.3 |
| | 10 | 6M | 8F | 0–0.5 | 0.3 |
| Mg-Dye-III $C_{17}H_{10}MgN_2O_3$ | 1 | 8MD | 4MD | 3–3.5 | 1 |
| | 3 | 6MD | 8F | 3–3.5 | 50 |
| | 5 | 8M | 4F | 3–3.5 | 3 |
| | 10 | 6M | 6M | 2–2.5 | 10 |
| Mg-Dye-IV $C_{25}H_{18}MgN_4O_6$ | 1 | 6F | 8F | 0.5–1 | 3 |
| | 3 | 6M | - | 1–1.5 | 3 |
| | 5 | 6M | - | 0–0.5 | 3 |
| | 10 | 8D | 8D | 0–0.5 | 1 |
| MgO | 1 | 8F | - | 0–0.5 | - |
| | 3 | 8F | 8F | 0.5–1 | - |
| | 5 | 8D | 8MD | 0.5–1 | 0.1 |
| | 10 | 8D | 8D | 0.5–1 | 0.1 |
| Ca-Mg-HPO$_4$ | 1 | 6MD | 8F | 0–0.5 | 1 |
| | 3 | 6M | 6F | 1–1.5 | 0.3 |
| | 5 | 6M | 6F | 0.5–1 | 0.3 |
| | 10 | 4F | 6MD | 0.5–1 | 0.3 |
| Zn | - | 4F | 8F | 1–1.5 | 3 |

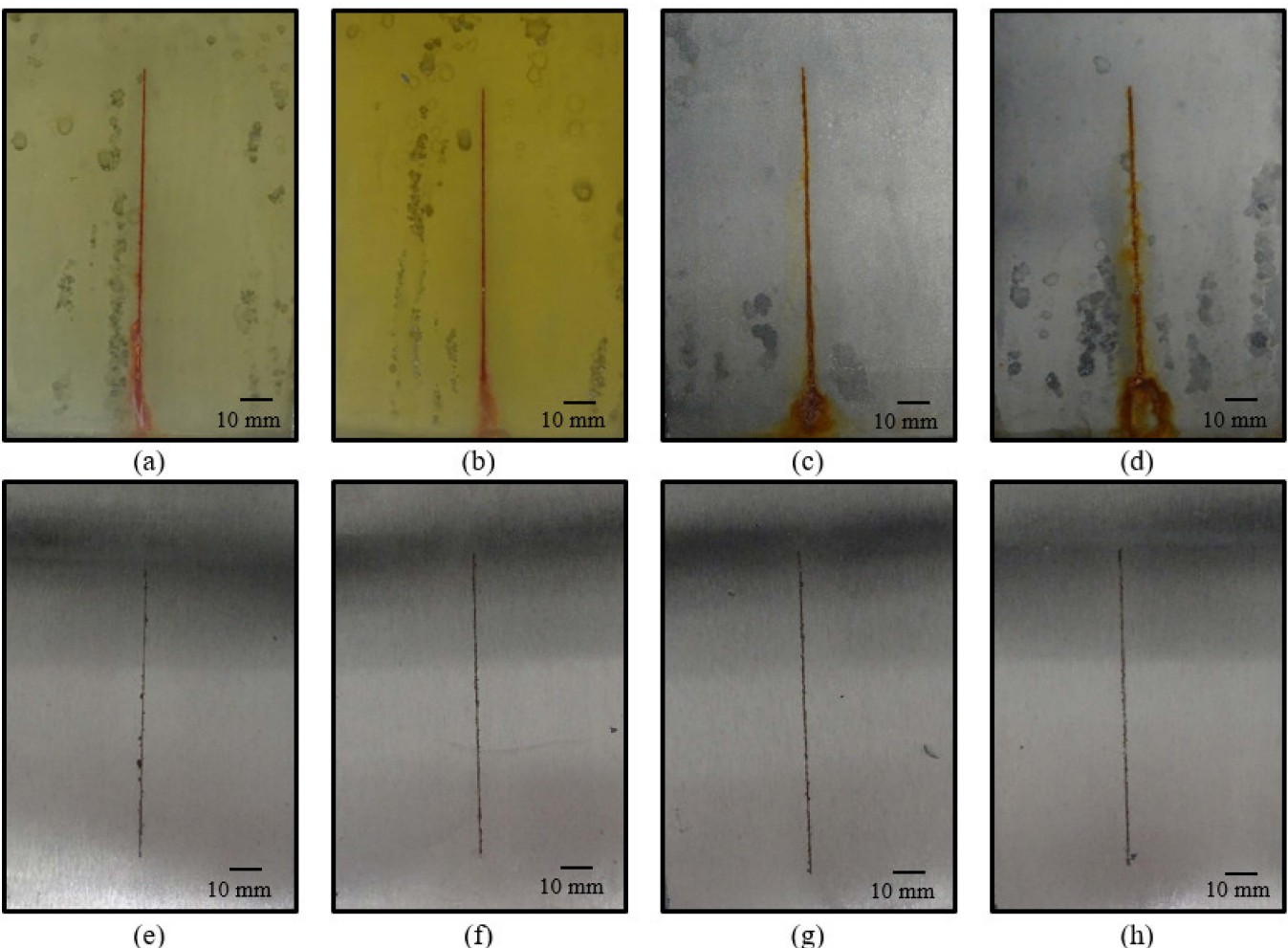

**Figure 17.** Organic coating after 1440 h of exposure in an atmosphere containing $SO_2$ and steel panel before removing the organic coating. Organic coating with: (**a**) $C_{34}H_{26}MgN_8O_6$ at PVC = 5%, (**b**) $C_{34}H_{26}MgN_8O_6$ at PVC = 10%, (**c**) MgO at PVC = 1%, (**d**) MgO at PVC = 3%; steel panel after removing the organic coating with: (**e**) $C_{34}H_{26}MgN_8O_6$ at PVC = 5%, (**f**) $C_{34}H_{26}MgN_8O_6$ at PVC = 10%, (**g**) MgO at PVC = 1%, (**h**) MgO at PVC = 3%.

With some exceptions, the investigated organic coatings showed good anti-corrosion efficiency. Compared to a corrosion test in a neutral salt mist atmosphere, the given coatings had a lower corrosion rate in the test cut and a smaller corrosion attack on the steel substrate. For all organic coatings, the dry thickness of the coating film was also reduced by about $15 \pm 5$ µm due to degradation processes [54]. Organic coatings with $C_{34}H_{26}MgN_8O_6$ pigment (Mg-Dye-I) achieved higher corrosion efficiency compared to organic coatings pigmented with other types of organic pigments. Of the organic coatings containing this type of pigment, the highest efficiency was achieved by the coating film at a PVC = 10%, in which blistering on the surface of the coating film was rated 8F, blistering in the test cut was rated 6F, corrosion in the vicinity of the test cut was also low (in the range of 0–0.5 mm), and corrosion attack on the steel substrate was low (0.1%). The organic coating with $C_{25}H_{18}MgN_4O_6$ pigment (Mg-Dye-IV) at a PVC = 5% also achieved a higher anti-corrosion efficiency, in which there were no blisters in the test cut, and the corrosion in the test cut was very low (in the range of 0–0.5 mm). In contrast, organic coatings with the pigment $C_{17}H_{10}MgN_2O_3$ (Mg-Dye-III) achieved low corrosion efficiency but achieved higher corrosion in the vicinity of the test cut (in the range of 2–3.5 mm) and a higher rate of blister formation, and the organic coating at a PVC = 3% even showed corrosion of the steel substrate from 50%. Of the organic coatings containing inorganic pigments, the coating

films with the pigment MgO at PVCs = 1 and 3% were effective, with no steel substrate corrosion, while blisters in the coating film area were classified as grade 8F, and corrosion in the test cut ranged from 0 to 1 mm.

### 4.8.3. Cyclic Corrosion/Weather Resistance Test with Exposure to a Salt Electrolyte (NaCl + (NH$_4$)$_2$SO$_4$) and UV Radiation

The organic coatings containing the tested pigments were also subjected to a cyclic corrosion and weather test for 1440 h, and the results are shown in Table 7. Photographs of the panels after 1440 h of exposure are presented in Figure 18.

**Table 7.** Results of cyclic corrosion testing in a fluorescent UV/in an atmosphere of NaCl + (NH$_4$)$_2$SO$_4$ with water steam condensation of the studied organic coatings containing studied pigments (PVC 1, 3, 5 and 10%) and zinc (PVC/CPVC = 0.60) after 1440 h of exposure, DFT = 90 ± 10 μm.

| Pigment | PVC [%] | Blistering | | Corrosion | |
|---|---|---|---|---|---|
| | | On the Film Area [dg] | In the Cut [dg] | In the Cut [mm] | Metal Base [%] |
| Mg-Dye-I C$_{34}$H$_{26}$MgN$_8$O$_6$ | 1 | 8F | 6M | 2–2.5 | 0.01 |
| | 3 | 8F | 8F | 2–2.5 | - |
| | 5 | 8F | 8M | 1.5–2 | - |
| | 10 | - | 8M | 1.5–2 | - |
| Mg-Dye-II C$_{26}$H$_{19}$MgN$_3$O$_5$ | 1 | 8M | 6M | 2.5–3 | 0.1 |
| | 3 | 6MD | 6M | 2.5–3 | 0.1 |
| | 5 | 6M | 6M | 2.5–3 | 0.1 |
| | 10 | 8F | 6MD | 2.5–3 | 0.1 |
| Mg-Dye-III C$_{17}$H$_{10}$MgN$_2$O$_3$ | 1 | 8MD | 6M | 2.5–3 | 0.1 |
| | 3 | 8D | 6M | 2.5–3 | 0.1 |
| | 5 | 8F | 6M | 2.5–3 | 0.1 |
| | 10 | 8F | 6M | 2.5–3 | 0.1 |
| Mg-Dye-IV C$_{25}$H$_{18}$MgN$_4$O$_6$ | 1 | 6F | 6F | 2.5–3 | 1 |
| | 3 | 8F | 6M | 2.5–3 | 0.01 |
| | 5 | 8M | 6M | 2.5–3 | 0.03 |
| | 10 | 8M | 6M | 2–2.5 | 0.1 |
| MgO | 1 | 8F | 6M | 2–2.5 | - |
| | 3 | 8F | 6M | 2–2.5 | - |
| | 5 | 8F | 8M | 2–2.5 | - |
| | 10 | 8F | 8M | 2–2.5 | - |
| Ca-Mg-HPO$_4$ | 1 | 8M | 8M | 2–2.5 | 0.1 |
| | 3 | 8F | 6M | 2.5–3 | 0.1 |
| | 5 | - | 6M | 2–2.5 | 0.1 |
| | 10 | 8F | 6M | 2–2.5 | 0.1 |
| Zn | - | 8M | 4M | 3–3.5 | 0.1 |

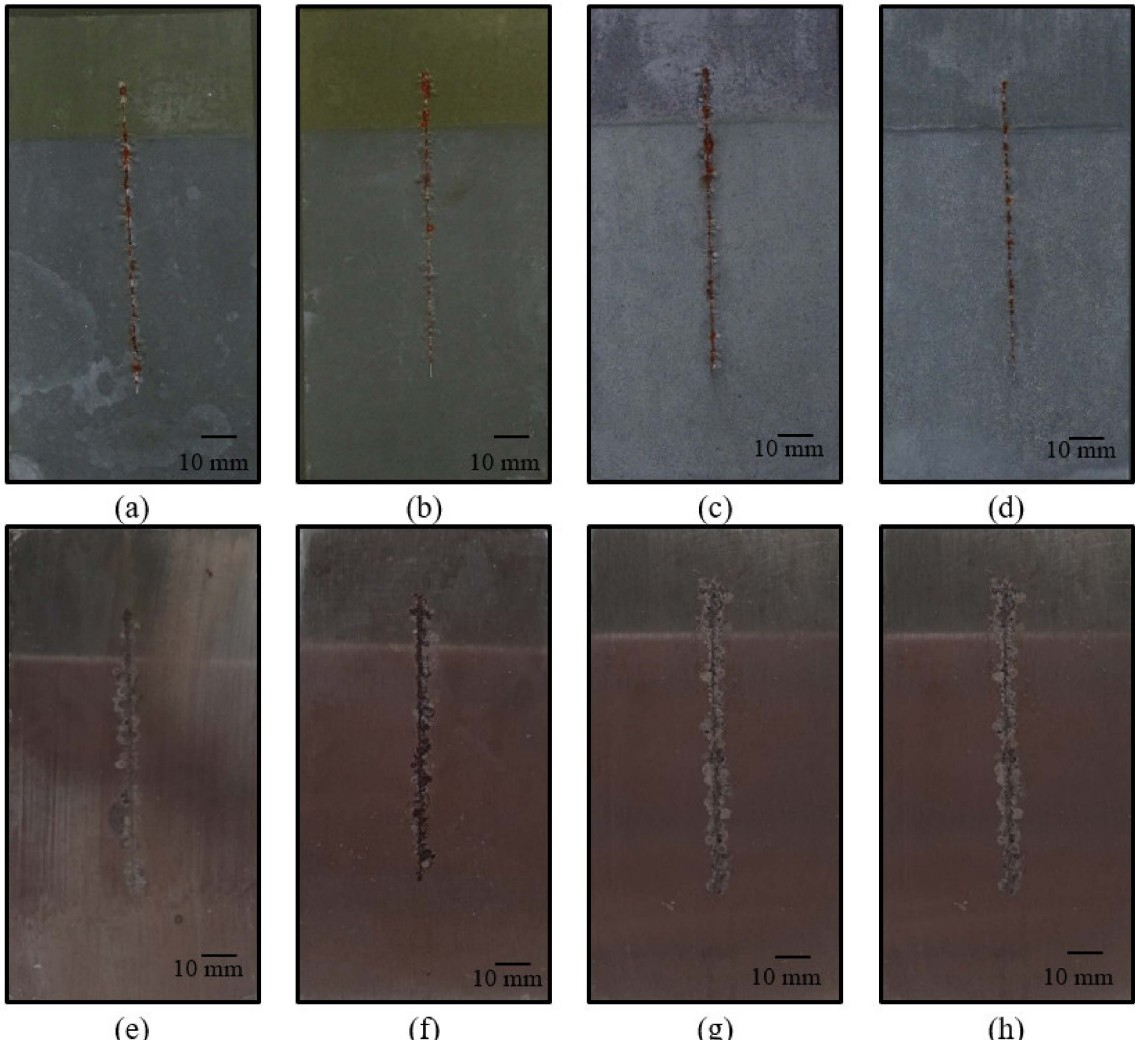

**Figure 18.** Organic coating after 1440 h of exposure to a salt electrolyte (NaCl + (NH$_4$)$_2$SO$_4$) and UV radiation and steel panel before removing the organic coating. Organic coating with: (**a**) C$_{34}$H$_{26}$MgN$_8$O$_6$ at PVC = 5%, (**b**) C$_{34}$H$_{26}$MgN$_8$O$_6$ at PVC = 10%, (**c**) MgO at PVC = 1%, (**d**) MgO at PVC = 3%; steel panel after removing the organic coating with: (**e**) C$_{34}$H$_{26}$MgN$_8$O$_6$ at PVC = 5%, (**f**) C$_{34}$H$_{26}$MgN$_8$O$_6$ at PVC = 10%, (**g**) MgO at PVC = 5%, (**h**) MgO at PVC = 10%.

The investigated organic coatings showed good corrosion resistance and high resistance to weather. All paint films showed little corrosion in the test cut (range 1.5–3.5 mm) and very little corrosion of the steel substrate (range 0–1%) after the paint film was removed. Among the organic coatings, coatings with the pigment C$_{34}$H$_{26}$MgN$_8$O$_6$ (Mg-Dye-I) achieved high efficiency, with no corrosion of the steel substrate at PVCs = 5 and 10%; corrosion in the test cut was also small (in the range of 1.5–2.0 mm), blisters in the test cut were rated 8M, and blisters in the coating film area were rated 8F. In contrast, organic coatings with the pigments C$_{26}$H$_{19}$MgN$_3$O$_5$ (Mg-Dye-II) and C$_{17}$H$_{10}$MgN$_2$O$_3$ (Mg-Dye-III) were less effective and caused more blistering in the coating film. Of the organic coatings containing inorganic pigments, the highest efficiency was achieved by the MgO-pigmented coatings, and their effectiveness against corrosion and weather also increased with increasing PVC values. All paint films with MgO pigment showed no corrosion of the steel substrate, the blisters in the area were graded 8F, the blisters in the test cut were graded 8M–6M, and the corrosion around the test cut was low (in the range of 2–2.5 mm).

### 4.9. Compositional Characterization of the Organic Coating after Exposure to the Salt Atmosphere

The studied organic coatings after 1440 h of exposure to the salt fog atmosphere were examined by scanning electron microscopy and energy dispersive spectroscopy. The organic coatings' composition in areas far from the test cut is shown in Figure 19, and the organic coatings' composition in a vicinity close to the test cut is shown on Figure 20.

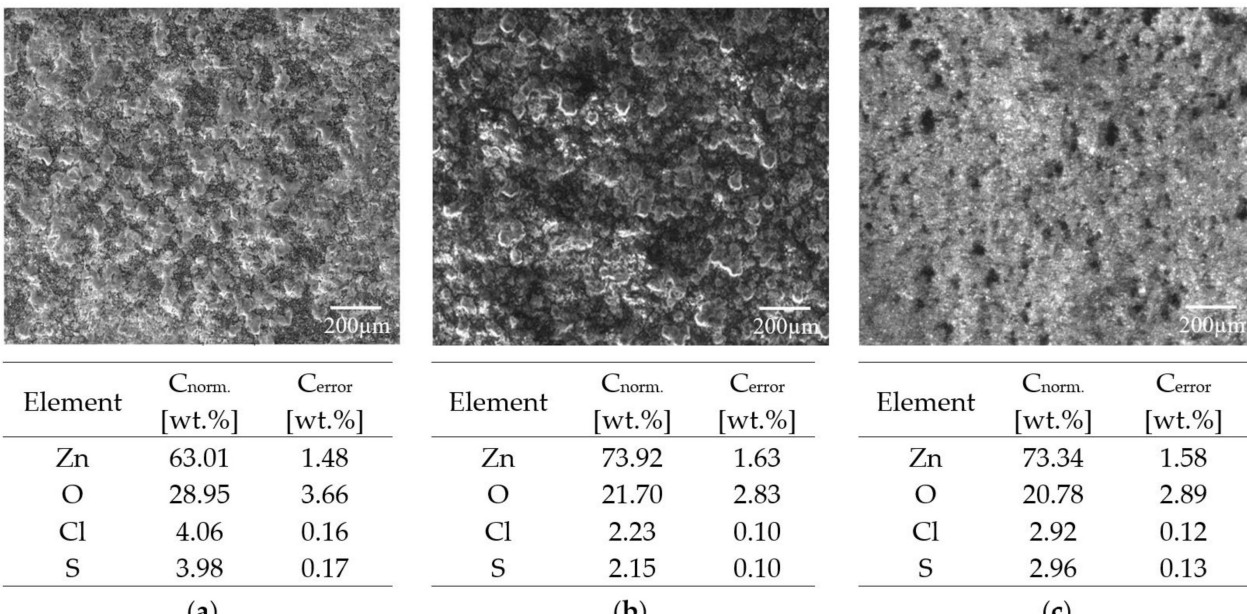

| Element | $C_{norm.}$ [wt.%] | $C_{error}$ [wt.%] |
|---------|--------------------|--------------------|
| Zn | 63.01 | 1.48 |
| O | 28.95 | 3.66 |
| Cl | 4.06 | 0.16 |
| S | 3.98 | 0.17 |

(**a**)

| Element | $C_{norm.}$ [wt.%] | $C_{error}$ [wt.%] |
|---------|--------------------|--------------------|
| Zn | 73.92 | 1.63 |
| O | 21.70 | 2.83 |
| Cl | 2.23 | 0.10 |
| S | 2.15 | 0.10 |

(**b**)

| Element | $C_{norm.}$ [wt.%] | $C_{error}$ [wt.%] |
|---------|--------------------|--------------------|
| Zn | 73.34 | 1.58 |
| O | 20.78 | 2.89 |
| Cl | 2.92 | 0.12 |
| S | 2.96 | 0.13 |

(**c**)

**Figure 19.** Results of scanning electron micrographs and energy-dispersive X-ray analysis of the organic coating in areas far from the test cut: (**a**) organic coating containing only zinc, (**b**) organic coating containing $C_{34}H_{26}MgN_8O_6$ (PVC = 10%) and zinc, (**c**) organic coating containing MgO (PVC = 10%) and zinc.

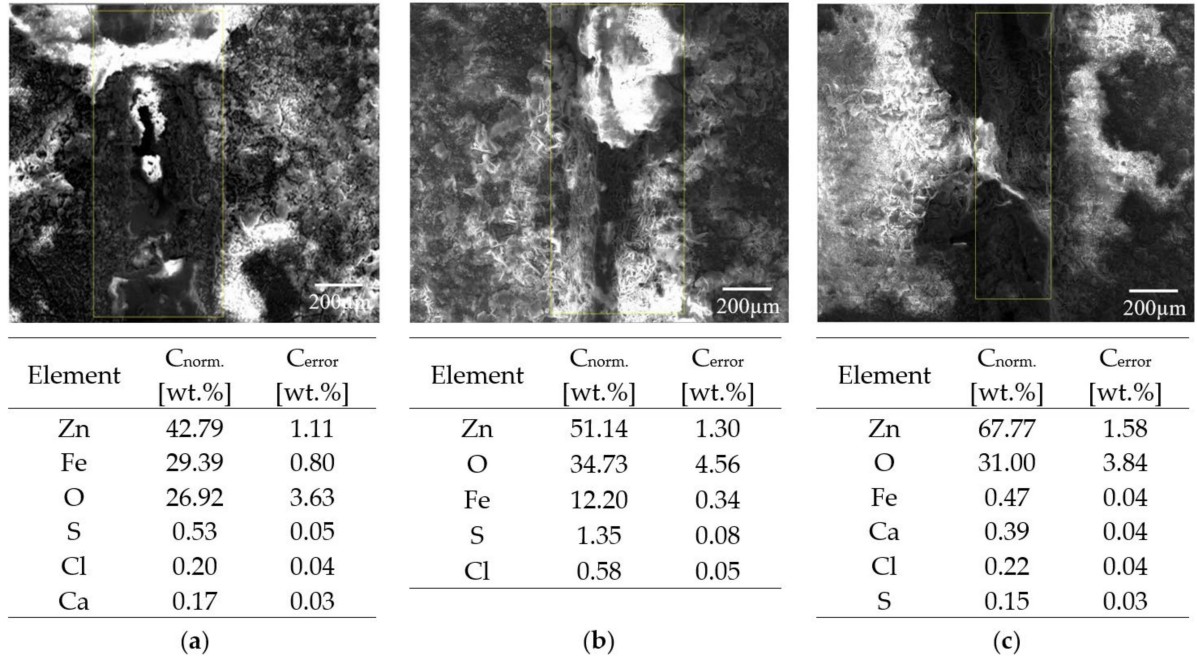

| Element | $C_{norm.}$ [wt.%] | $C_{error}$ [wt.%] |
|---------|--------------------|--------------------|
| Zn | 42.79 | 1.11 |
| Fe | 29.39 | 0.80 |
| O | 26.92 | 3.63 |
| S | 0.53 | 0.05 |
| Cl | 0.20 | 0.04 |
| Ca | 0.17 | 0.03 |

(**a**)

| Element | $C_{norm.}$ [wt.%] | $C_{error}$ [wt.%] |
|---------|--------------------|--------------------|
| Zn | 51.14 | 1.30 |
| O | 34.73 | 4.56 |
| Fe | 12.20 | 0.34 |
| S | 1.35 | 0.08 |
| Cl | 0.58 | 0.05 |

(**b**)

| Element | $C_{norm.}$ [wt.%] | $C_{error}$ [wt.%] |
|---------|--------------------|--------------------|
| Zn | 67.77 | 1.58 |
| O | 31.00 | 3.84 |
| Fe | 0.47 | 0.04 |
| Ca | 0.39 | 0.04 |
| Cl | 0.22 | 0.04 |
| S | 0.15 | 0.03 |

(**c**)

**Figure 20.** Results of scanning electron micrographs and energy-dispersive X-ray analysis of the organic coating in close vicinity to the test cut: (**a**) organic coating containing only zinc, (**b**) organic coating containing $C_{34}H_{26}MgN_8O_6$ (PVC = 10%) and zinc, (**c**) organic coating containing MgO (PVC = 10%) and zinc.

X-ray diffraction analysis (XRD) of the zinc particles provided evidence that zinc had been present in the paint in the metal dust form before the corrosion test. Energy dispersive spectroscopy (Figure 20) showed that the pores in the organic coatings containing zinc had been sealed by the zinc corrosion products (ZnO). This conclusion was confirmed by the XRD data. Steel corrosion products were not found on organic coatings containing zinc after 1440 h of exposure to the salt atmosphere, attesting to a high degree of corrosion protection provided by the studied organic coatings. This fact applies to all organic coatings studied in this work.

Results of energy dispersive spectroscopy (Figure 20) demonstrated that the test cuts prepared in organic coatings were partly sealed by the zinc corrosion products (ZnO) due to the effects of the cathodic protection mechanism. The steel corrosion products were also found in a vicinity close to the test cut after 1440 h of exposure to the salt atmosphere. The presence of these products indicated that the cathodic protection mechanism was not rapid enough to prevent corrosion of the steel substrate. The results of energy dispersive spectroscopy are shown in Figure 20. From these results, it is clear that, in the case of the organic coating containing $C_{34}H_{26}MgN_8O_6$ (Mg-Dye-I) particles (PVC = 10%) and zinc particles or magnesium oxide particles (PVC = 10%) and zinc particles, electrochemical protection was enhanced compared to the standard organic coating containing only spherical zinc particles. The results of energy dispersive spectroscopy are in line with the conclusions of cyclic corrosion testing in an atmosphere of NaCl with water steam condensation. For the organic coating containing $C_{34}H_{26}MgN_8O_6$ particles (PVC = 10%) and zinc particles, corrosion in the cross-section was achieved for a distance of only 1.5–2 mm, and for the organic coating containing MgO particles (PVC = 10%) and zinc particles, this corrosion parameter achieved a value of 2–2.5 mm, whereas in the case of the standard organic coating containing only spherical Zn particles, the corrosion in the cross-section was extended to a distance of 8–8.5 mm.

### 4.10. Potentiodynamic Polarization Studies and Electrochemical Impedance Spectroscopy

The results of potentiodynamic polarization studies and electrochemical impedance spectroscopy are shown in Table 8. Figure 21 shows Tafel plots of selected studied organic coatings, which are discussed below. The $C_{34}H_{26}MgN_8O_6$ (Mg-Dye-I) pigmented coating exhibited excellent corrosion resistance properties compared to other type of coating. In this group of coatings, 10% PVC exhibited the lowest corrosion rate of 0.92 mpy and the highest impedance values. The organic coatings containing $C_{25}H_{18}MgN_4O$ (Mg-Dye-IV) and the coatings having MgO as pigments performed moderately among all types of coatings, whereas the coatings with the pigments $C_{26}H_{19}MgN_3O_5$ (Mg-Dye-II), Ca-Mg-HPO$_4$, and $C_{17}H_{10}MgN_2O_3$ (Mg-Dye-III) revealed higher corrosion rates and lower impedance values. In the class of MgO coatings, the coatings having 1% and 3% PVCs were more effective than the other coatings. These results support accelerated corrosion testing in a salt mist atmosphere, accelerated corrosion testing in atmosphere containing SO$_2$, cyclic corrosion, and weather resistance testing (Tables 5–7).

**Table 8.** Results of potentiodynamic polarization studies and electrochemical impedance spectroscopy of organic coatings containing studied pigments (PVC 1, 3, 5 or 10%) and zinc (PVC/CPVC = 0.60). DFT = 50 ± 5 μm.

| Pigment | PVC [%] | $E_{corr}$ [mV] | $I_{corr}$ [$\mu A \cdot cm^{-2}$] | Corrosion Rate [mpy] | $Z_{mod}$ [$\Omega \cdot cm^{-2}$] | $Z_{real}$ [$\Omega \cdot cm^{-2}$] |
|---|---|---|---|---|---|---|
| | 1 | −748 | 19.6 | 1.86 | 134.4 | 113.7 |
| Mg-Dye-I | 3 | −755 | 10.2 | 0.97 | 151.1 | 133.7 |
| $C_{34}H_{26}MgN_8O_6$ | 5 | −726 | 18.1 | 1.71 | 113.7 | 98.01 |
| | 10 | −703 | 9.69 | 0.92 | 197.4 | 183.5 |

**Table 8.** *Cont.*

| Pigment | PVC [%] | $E_{corr}$ [mV] | $I_{corr}$ [$\mu A \cdot cm^{-2}$] | Corrosion Rate [mpy] | $Z_{mod}$ [$\Omega \cdot cm^{-2}$] | $Z_{real}$ [$\Omega \cdot cm^{-2}$] |
|---|---|---|---|---|---|---|
| Mg-Dye-II $C_{26}H_{19}MgN_3O_5$ | 1 | −722 | 43.1 | 4.10 | 57.26 | 31.77 |
| | 3 | −716 | 51.1 | 4.90 | 31.70 | 17.82 |
| | 5 | −721 | 42.7 | 4.04 | 74.00 | 61.40 |
| | 10 | −672 | 44.2 | 4.19 | 45.82 | 23.44 |
| Mg-Dye-III $C_{17}H_{10}MgN_2O_3$ | 1 | −694 | 72.3 | 6.84 | 25.09 | 11.91 |
| | 3 | −714 | 79.4 | 7.52 | 29.33 | 15.68 |
| | 5 | −700 | 83.9 | 7.94 | 22.19 | 13.37 |
| | 10 | −710 | 91.6 | 8.37 | 25.94 | 14.16 |
| Mg-Dye-IV $C_{25}H_{18}MgN_4O_6$ | 1 | −719 | 41.5 | 3.92 | 64.02 | 42.86 |
| | 3 | −703 | 35.1 | 3.33 | 74.00 | 49.25 |
| | 5 | −726 | 32.6 | 3.10 | 90.45 | 59.08 |
| | 10 | −687 | 48.0 | 4.54 | 69.21 | 46.20 |
| MgO | 1 | −776 | 10.3 | 0.97 | 151.1 | 133.7 |
| | 3 | −718 | 13.3 | 1.26 | 111.8 | 98.01 |
| | 5 | −732 | 14.7 | 1.39 | 54.76 | 34.87 |
| | 10 | −720 | 15.3 | 1.45 | 66.94 | 46.37 |
| Ca-Mg-HPO$_4$ | 1 | −719 | 49.3 | 4.67 | 59.21 | 28.38 |
| | 3 | −687 | 51.7 | 4.89 | 43.82 | 24.31 |
| | 5 | −700 | 49.2 | 4.66 | 54.76 | 28.47 |
| | 10 | −716 | 51.1 | 4.86 | 40.08 | 23.37 |
| Zn | - | −737 | 23.8 | 2.25 | 22.44 | 11.38 |

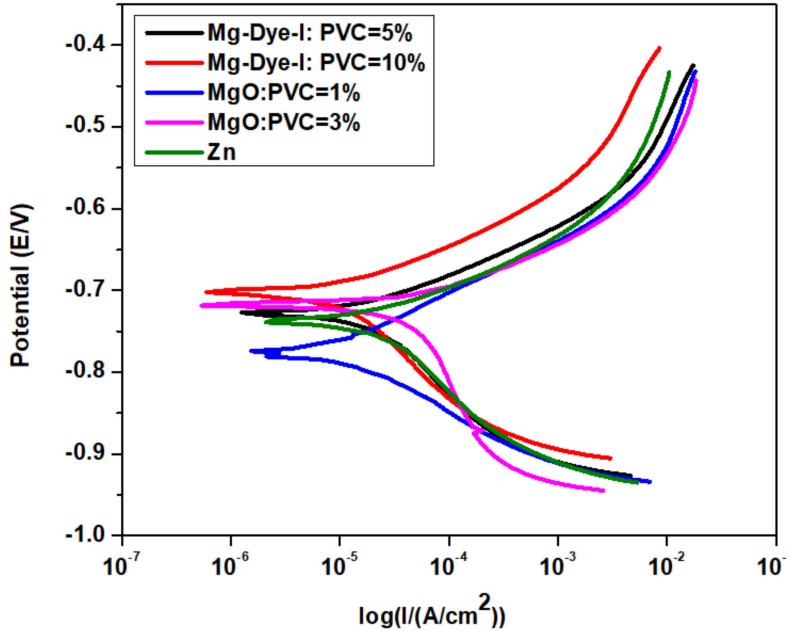

**Figure 21.** Tafel plots of studied organic coatings containing $C_{34}H_{26}MgN_8O_6$ (PVC = 5 and 10%), MgO (PVC = 1 and 3%), and the standard organic coating pigmented only with zinc (PVC/CPVC = 0.60).

*4.11. Determination of pH and Specific Electrical Conductivity and Corrosion Loss from Aqueous Extracts of Pigments and of Loose Paint Films*

For these determinations, suspensions (10 wt.%) were prepared in redistilled water with an initial pH = 6.99 and specific electrical conductivity ($\lambda$) = 1.5 $\mu S \cdot cm^{-1}$. The pH values and specific electrical conductivity of the prepared suspensions were subsequently measured over the course of 28 days. The measurement results related to the pigment

extracts are shown in Table 9, while the results related to the loose paint film extracts are shown in Table 10.

**Table 9.** The values of $pH_p$ and specific electrical conductivity ($\lambda_p$) of 10% suspensions of studied pigments (measured at 1 and 28 days) and corrosion losses in aqueous extracts of these pigments.

| Pigment | [a] $pH_p^1$ | [a] $pH_p^{28}$ | [b] $\lambda_p^1$ [$\mu S \cdot cm^{-1}$] | [b] $\lambda_p^{28}$ [$\mu S \cdot cm^{-1}$] | $K_{mp}$ [$g \cdot m^{-2}$] | $X_{Hp}$ [%] | $U_{Rp}$ [mm] | $V_{Kp}$ [$g \cdot m^{-2} \cdot d^{-1}$] |
|---|---|---|---|---|---|---|---|---|
| Mg-Dye-I $C_{34}H_{26}MgN_8O_6$ | 10.28 | 10.48 | 449 | 468 | 0.471 | 34.6 | $6.0 \times 10^{-5}$ | $6.7 \times 10^{-2}$ |
| Mg-Dye-II $C_{26}H_{19}MgN_3O_5$ | 10.45 | 10.54 | 301 | 523 | 0.819 | 60.2 | $1.0 \times 10^{-4}$ | $1.2 \times 10^{-1}$ |
| Mg-Dye-III $C_{17}H_{10}MgN_2O_3$ | 10.55 | 10.98 | 890 | 911 | 1.265 | 92.9 | $1.6 \times 10^{-4}$ | $1.8 \times 10^{-1}$ |
| Mg-Dye-IV $C_{25}H_{18}MgN_4O_6$ | 10.33 | 10.53 | 428 | 521 | 1.262 | 92.7 | $1.6 \times 10^{-4}$ | $1.8 \times 10^{-1}$ |
| MgO | 10.23 | 10.32 | 640 | 854 | 0.597 | 43.9 | $7.6 \times 10^{-5}$ | $8.5 \times 10^{-2}$ |
| Ca-Mg-HPO$_4$ | 6.41 | 6.51 | 980 | 998 | 1.139 | 83.7 | $1.5 \times 10^{-4}$ | $1.6 \times 10^{-1}$ |
| H$_2$O (distilled) | 7.19 | 7.10 | 50 | 59 | 1.361 | 100 | $1.7 \times 10^{-4}$ | $1.9 \times 10^{-1}$ |

[a] pH was measured with an accuracy of $\pm0.01$. [b] Specific electric conductivity was measured with an accuracy of $\pm0.5\%$. Parameters are reported as arithmetic averages of five measured values.

**Table 10.** The values of $pH_f$ and specific electrical conductivity ($\lambda_f$) of 10% suspensions of studied loose paint films (measured at 1 and 28 days) and corrosion losses in aqueous extracts of these loose paint films.

| Pigment | $PVC_{pigm}$ [%] | [a] $pH_f^1$ | [a] $pH_f^{28}$ | [b] $\lambda_f^1$ [$\mu S \cdot cm^{-1}$] | [b] $\lambda_f^{28}$ [$\mu S \cdot cm^{-1}$] | $K_{Mf}$ [$g \cdot m^{-2}$] | $X_{Hf}$ [%] | $U_{Rf}$ [mm] | $V_{Kf}$ [$g \cdot m^{-2} \cdot d^{-1}$] |
|---|---|---|---|---|---|---|---|---|---|
| Mg-Dye-I | 1 | 6.41 | 6.80 | 26 | 55 | 0.695 | 51.0 | $8.9 \times 10^{-5}$ | $9.9 \times 10^{-2}$ |
| $C_{34}H_{26}MgN_8O_6$ | 10 | 6.74 | 6.95 | 64 | 72 | 0.587 | 43.1 | $7.5 \times 10^{-5}$ | $8.4 \times 10^{-2}$ |
| Mg-Dye-II | 1 | 6.22 | 6.58 | 44 | 58 | 1.111 | 74.2 | $1.4 \times 10^{-4}$ | $1.6 \times 10^{-1}$ |
| $C_{26}H_{19}MgN_3O_5$ | 10 | 6.33 | 6.86 | 46 | 69 | 0.945 | 69.3 | $1.2 \times 10^{-4}$ | $1.4 \times 10^{-1}$ |
| Mg-Dye-III | 1 | 6.48 | 6.62 | 40 | 61 | 1.354 | 99.3 | $1.7 \times 10^{-4}$ | $1.9 \times 10^{-1}$ |
| $C_{17}H_{10}MgN_2O_3$ | 10 | 6.58 | 6.80 | 47 | 77 | 1.289 | 96.4 | $1.6 \times 10^{-4}$ | $1.8 \times 10^{-1}$ |
| Mg-Dye-IV | 1 | 6.12 | 6.67 | 43 | 92 | 1.359 | 99.7 | $1.7 \times 10^{-4}$ | $1.9 \times 10^{-1}$ |
| $C_{25}H_{18}MgN_4O_6$ | 10 | 6.39 | 6.92 | 56 | 95 | 1.291 | 96.7 | $1.6 \times 10^{-4}$ | $1.8 \times 10^{-1}$ |
| MgO | 1 | 6.81 | 6.92 | 60 | 96 | 0.696 | 50.7 | $8.9 \times 10^{-5}$ | $9.9 \times 10^{-2}$ |
| | 10 | 7.27 | 7.42 | 96 | 101 | 0.678 | 49.7 | $8.6 \times 10^{-5}$ | $9.7 \times 10^{-2}$ |
| Ca-Mg-HPO$_4$ | 1 | 6.86 | 6.66 | 55 | 102 | 1.347 | 98.8 | $1.7 \times 10^{-4}$ | $1.9 \times 10^{-1}$ |
| | 10 | 6.64 | 6.55 | 89 | 150 | 1.245 | 91.3 | $1.6 \times 10^{-4}$ | $1.8 \times 10^{-1}$ |
| Zn | - | 6.83 | 6.56 | 40 | 83 | 1.215 | 89.1 | $1.5 \times 10^{-4}$ | $1.7 \times 10^{-1}$ |
| non-pigmented film | - | 5.75 | 5.55 | 48 | 53 | 1.236 | 90.7 | $1.6 \times 10^{-4}$ | $1.8 \times 10^{-1}$ |
| H$_2$O (distilled) | - | 7.18 | 7.09 | 49 | 60 | 1.363 | 100 | $1.7 \times 10^{-4}$ | $1.9 \times 10^{-1}$ |

[a] pH was measured with an accuracy of $\pm 0.01$. [b] Specific electric conductivity was measured with an accuracy of $\pm0.5\%$. Parameters are reported as arithmetic averages of five measured values.

From the measured results ($pH_p^1$ and $pH_p^{28}$), it can be concluded that the extracts from the studied organic pigments and the MgO pigment had alkaline pH values. These alkaline pH values of the above-mentioned organic pigment extracts (10.48–10.98) are mainly caused by the release of Mg$^{2+}$ ions present in the pigments in question. These Mg$^{2+}$ ions can subsequently change to magnesium compounds (Mg(OH)$_2$ and MgO) when in contact with a metal substrate in an aqueous environment (Figure 22). The alkaline pH values of the MgO pigment extract (10.32) are caused by the limited solubility of MgO in water, resulting in the formation of alkaline Mg(OH)$_2$, which is also why MgO is referred to by the manufacturer (on the technical datasheet) as a mild basic reagent. Acidic pH values of the extract (6.51) were recorded only when studying the Ca-Mg-HPO$_4$ pigment

due to the release of an acidic phosphate anion. The release of the above-mentioned ions into the aqueous extracts resulted in an increase in specific electrical conductivity values by two orders of magnitude. The highest values of specific electrical conductivity ($998\ \mu S \cdot cm^{-1}$) were achieved by the $Ca\text{-}Mg\text{-}HPO_4$ pigment extract compared to the other types of studied pigments. Determination of corrosion losses from aqueous pigment extracts ($K_{Mp}$) was used to determine mass and dimensional changes in steel panels ($X_{Hp}$, $U_{Rp}$ and $V_{Kp}$). For this determination, extracts prepared for determining the pH value and specific electrical conductivity of the studied pigments were used when steel panels were immersed in these extracts for a period of 7 days. The lowest weight loss was recorded in the extract of the organic pigment $C_{34}H_{26}MgN_8O_6$ (Mg-Dye-I) ($K_{Mp}$ = 0.471 $g.m^{-2}$) when the weight losses of steel ($V_{Kp}$ = 6.7 × $10^{-2}$ $g \cdot m^{-2} \cdot d^{-1}$) also reached an order of magnitude lower value compared to the other types of studied organic pigments (1.2 × $10^{-1}$–1.8 × $10^{-1}$ $g \cdot m^{-2} \cdot d^{-1}$). Similar values of the evaluated parameters $K_{Mp}$ and $V_{Kp}$ (compared to the pigment extract $C_{34}H_{26}MgN_8O_6$) were recorded in the MgO pigment extract. Based on these results, it can be assumed that these two types of pigments released a greater number of inhibitory ions into the aqueous environment compared to the other studied pigments.

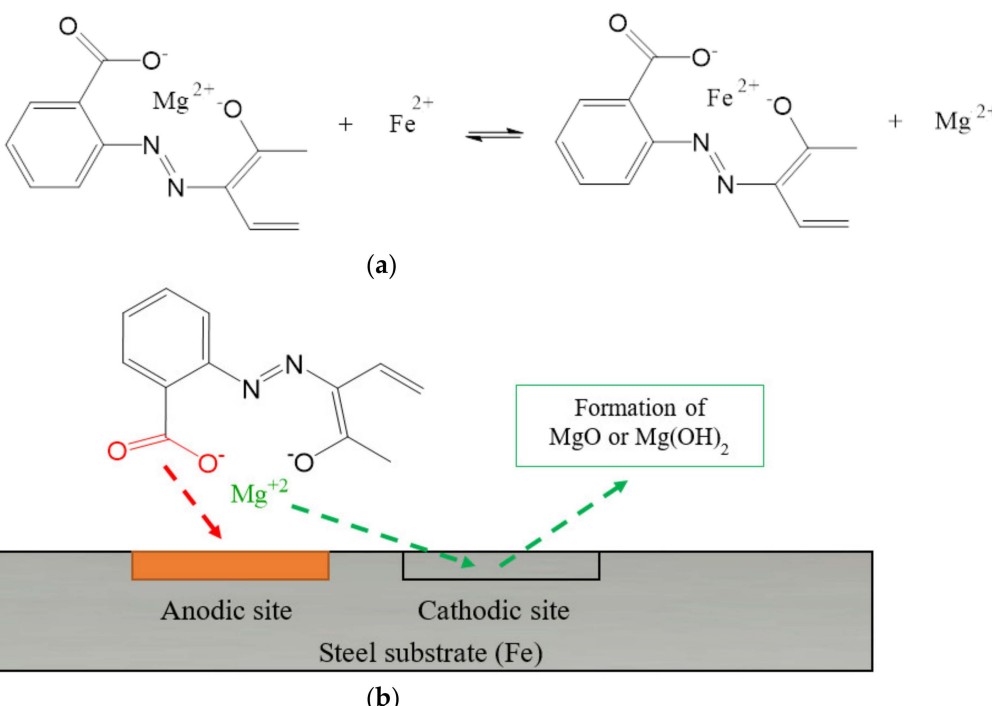

(a)

(b)

**Figure 22.** (**a**,**b**) Schematic representation of the protective effect of pigments in powder form on a metal surface in contact with water (without the presence of a binder component).

From the measured results ($pH_f^1$ and $pH_f^{28}$), it can be concluded that the extracts of loose paint films reached lower pH values compared to the aqueous extracts of the studied organic pigments ($pH_p^1$ and $pH_p^{28}$). A similar dependence can also be seen from the results of determining the specific electrical conductivity, when the extracts of free films ($\lambda_f^1$ and $\lambda_f^{28}$) reached lower values compared to the extracts of the studied organic pigments ($\lambda_p^1$ and $\lambda_p^{28}$). The pH value and specific electrical conductivity in the paint film extracts were significantly influenced by the film-forming component, which outweighed the influence of the pigments fixed in it. This conclusion is confirmed by the non-pigmented film reaching the most acidic $pH_f^{28}$ = 5.55, when such an acidic pH can be explained by the acidic nature of the given type of binder [52]. Determination of corrosion losses from aqueous solutions of loose paint films extracts ($K_{Mp}$) was used to determine mass and dimensional changes in steel panels ($X_{Hf}$, $U_{Rf}$ and $V_{Kf}$). For this determination, extracts prepared for determining

the pH value and specific electrical conductivity of loose paint films were used when steel panels were immersed in these extracts for a period of 7 days. After this determination, the lowest weight losses ($K_{Mf}$) were recorded in the extracts of loose paint films containing the pigments $C_{34}H_{26}MgN_8O_6$ (Mg-Dye-I) and MgO, and this conclusion corresponds to the results of the determination of weight losses in the pigment extracts. This fact confirms the inhibitory abilities of the ions released from these two types of pigments, as the corrosion losses in the extracts of non-pigmented paint film reached higher values. The processes occurring in the cross-linked pigmented epoxy coating film in contact with water are shown schematically in Figure 23.

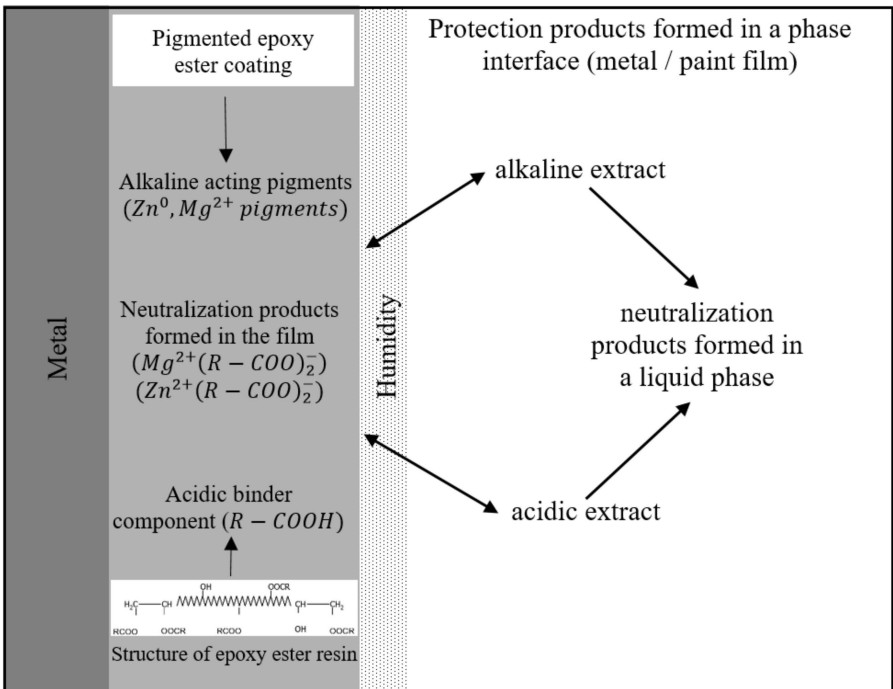

**Figure 23.** Schematic representation of interactions running in epoxyester-based paint film containing $Zn^0$ and $Mg^{2+}$ pigments in the presence of water [52].

### 4.12. Assumed Corrosion Protection Mechanism of Studied Organic Pigment in the Organic Coatings Containing Zinc Pigment

Anti-corrosion pigments can be defined in terms of their function as corrosion inhibitors as substances that, when they come in contact with the protected metal and in the presence of water and oxygen, slow the corrosion of the metal under the organic coating and actively participate in extending the life of the paint film. Anti-corrosion pigments or corrosion inhibitors do not directly reduce the concentration of corrosive substances but change the properties of the phase interface. Pigments can provide anti-corrosion action in binders and protective organic coatings by several mechanisms. Usually, there are three basic types of action mechanisms: barrier, chemical, and electrochemical mechanisms [36,45,55].

In the case of chemically acting inhibitors and pigments, the protective effect is related to the ability to create a layer of corrosion products on the surface of the metal, which limits the speed of the corrosion process. The protective layer can be formed by several means: oxidation of the metal, oxidation of the primary products of metal corrosion, reaction of the inhibitor with metal or with products of metal corrosion, and increasing the alkalinity of the environment in the zone of its immediate contact with the metal surface. Inhibition by the formation of a protective film is the most important function of common anti-corrosion pigments. If the pigment reduces the potential difference of the existing local cells through chemical and electrochemical influences so strongly that low current will flow between the cathode and the anode, the relevant pigment is considered to be optimal in terms of electrochemical performance. Pigments, which are electrochemically less noble than iron,

i.e., with a greater negative potential than $Fe/Fe^{2+}$, primarily provide cathodic protection and cause the formation of barriers and chemically acting oxidation products, which also limit the rate of corrosion reactions in the long term, leading to the degradation of the protective coating (for example, zinc dust, magnesium, lead). For application in binders of paints, these pigments and coatings have certain limitations in terms of preparation and effectiveness, so they cannot be compared with classic metal coatings, for example, the galvanic type [36,45,55].

The mechanism of zinc dust in classic binders for paint materials is based on the sacrificial anode principle, similar to galvanic coatings, and it is complicated for many reasons. A high-volume concentration of zinc dust would be required to enable this mechanism in the paint film. However, barrier and chemical mechanisms are important for anti-corrosion properties of organic coatings. The alkaline action of corrosion products ($Zn^{2+}$) and metallic zinc ($Zn^0$), and their conductive connection through the electrolyte in the zinc-pigmented paint film are thus sufficient to ensure anti-corrosion protection, even at a lower $PVC_{Zn}$ than $CPVC_{Zn}$, approx. from $PVC_{Zn} = 50\%$. Here, in this case, the coating containing zinc dust is porous (PVC = 50%), so the $Zn^0$ particles present in the coating film come in contact with the corrosive environment, and the metal substrate changes to $Zn^{2+}$ (Figure 24) [56,57].

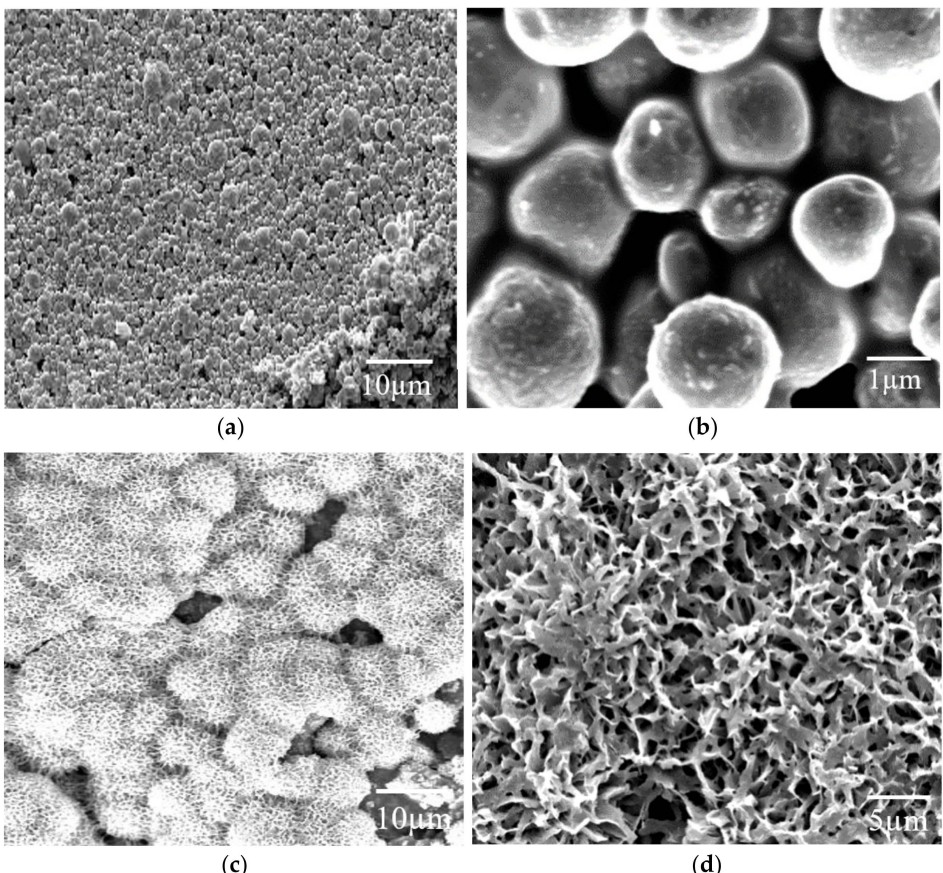

**Figure 24.** (**a**,**b**) Micrograph of a zinc-pigmented organic coating at PVC/CPVC = 0.60, (**c**,**d**) zinc-pigmented organic coating with zinc corrosion products after exposure of the coating to a corrosive environment.

Since the reactions of the zinc–steel corrosion cell near the steel surface are accompanied by an increasing pH in water that has diffused through the coating, film-forming substances that are also resistant at elevated pH levels must be selected for zinc-containing contact-type coating materials. Epoxy resins, polyurethanes, silicates, ethylsilicates, and epoxyester resins (specifically used in this study) are feasible binders.

For anti-corrosion coatings with Zn content, it is advantageous to include other inhibitors and pigments for the anti-corrosion function of the coating when the requirement for the mechanical resistance of the resulting organic coating is also higher. Depending on their type, these pigments can strengthen the chemical, barrier and electrochemical action of zinc dust or create a synergistic effect (zinc and anti-corrosion pigment). These pigments, which are present in the formulation of the zinc-containing coating material, should have a concentration in themselves that exhibits an anti-corrosion effect, even at a lower concentration than that of zinc dust itself [58].

These polymers are conductive, for example, graphite, carbon nanotubes, and other non-noble metals (magnesium), providing a synergistic effect in anti-corrosion efficiency when used along with zinc [58]. This publication specifically addresses organic and inorganic pigments that contain an alkaline-acting magnesium cation in their structure. The tested coatings with Zn dust content (PVC/CPVC = 0.60), in which $Zn^0$ particles partially change to $Zn^{2+}$, are alkaline in nature and also contain newly synthesized alkaline-acting pigments (Table 9). In this way, a greater number of alkaline-acting substances in the coating film are ensured, resulting in a positive effect in inhibiting the corrosion of the metal substrate (Figure 25).

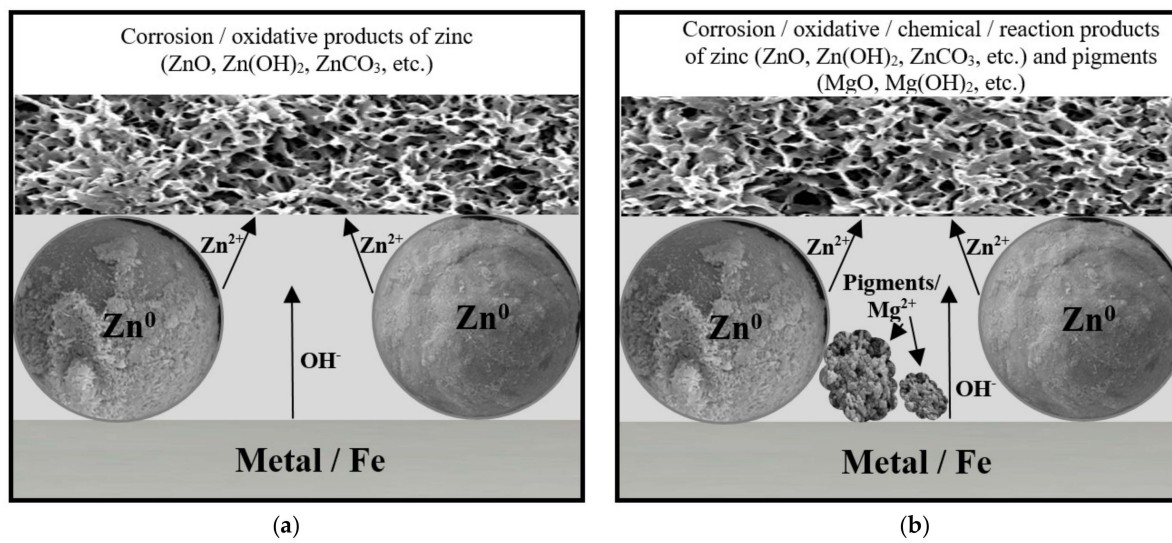

**Figure 25.** (**a**) Formation of corrosion products in a zinc-pigmented coating, (**b**) formation of corrosion products in a zinc-pigmented organic coating that additionally contained magnesium-containing pigments.

Weight loss of metal caused by corrosion in an aqueous environment (coating exposed to moisture) can be suppressed to a large extent by choosing a suitable pigment from which ions are released into the aqueous environment that decrease corrosion of the metal substrate or form passivation layers with iron ions. With the help of weight loss, it can then be determined whether the given pigment is suitable for application in paint as an anti-corrosion pigment or not [52].

The method of following the corrosion losses in the suspensions of organic coatings containing anticorrosive pigments gives evidence of the possible reactions inside the coating, both in the liquid state on the formation of film and also partially on aging the film in the hardened state (Table 10). This process allows for presuming that alkaline anticorrosive pigments, in this case the pigments $C_{34}H_{26}MgN_8O_6$ (Mg-Dye-I), $C_{26}H_{19}MgN_3O_5$ (Mg-Dye-II), $C_{17}H_{10}MgN_2O_3$ (Mg-Dye-III), $C_{25}H_{18}MgN_4O_6$ (Mg-Dye-IV), and MgO, can neutralize the acidic binder groups. The carboxyl groups contained in the epoxyester-based binder cause the acidity. It was found that the film, which does not contain any pigments, and which was transferred to the aqueous suspension, affects the water extract pH value in such a way that the resulting pH value is 5.55. When this acidic binder contains alka-

line pigment ($C_{34}H_{26}MgN_8O_6$ (Mg-Dye-I), $C_{26}H_{19}MgN_3O_5$ (Mg-Dye-II), $C_{17}H_{10}MgN_2O_3$ (Mg-Dye-III), $C_{25}H_{18}MgN_4O_6$ (Mg-Dye-IV)), or MgO), then the pH extract values increased with most pigments used to a range of pH values from 6.58 (coating with $C_{26}H_{19}MgN_3O_5$ (Mg-Dye-II)) to 7.42 (coating with MgO). This growth of pH value is caused by the formation of metal soaps in the film. In principle, the effect is similar to that of inhibitive red lead action [52]. The alkaline medium is not suitable for corrosion, unlike acidic medium. Metal soaps can also exhibit inhibitive properties at the protected metal/organic coating interface.

## 5. Conclusions

The effect of synthesized novel magnesium complex dyes in zinc pigmented organic coatings based on epoxyester resin was studied in this work. Four types of different magnesium complexes (Mg-Dye-I ($C_{34}H_{26}MgN_8O_6$), Mg-Dye-II ($C_{26}H_{19}MgN_3O_5$), Mg-Dye-III ($C_{17}H_{10}MgN_2O_3$), and Mg-Dye-IV ($C_{25}H_{18}MgN_4O_6$)) were synthesized and subsequently were characterized using a number of analytical methods (ICP-OES, EDX, HRMS-MALDI, XRD, UV-Vis and IR). Together with four types of different magnesium complexes, selected inorganic pigments (MgO, Ca-Mg-HPO$_4$) were subjected to measurements of the typical paint parameters (density, oil number) and the calculation of the critical pigment volume concentration and other necessary measurements, including SEM, SEM-EDX, and XRD. Subsequently, model coatings containing individual types of studied pigments (organic and inorganic types) at different values of PVC (1, 3, 5, and 10%) and zinc (at PVC/CPVC = 0.60) were formulated and prepared by the dispersion process. Prepared model paint materials were applied to steel panels, and then the effect of individual types of pigments on the corrosion resistance behaviour of coatings was studied after exposure to a corrosive environment. The corrosion resistance of organic coatings was tested in accelerated corrosion tests in a salt mist atmosphere, in an atmosphere containing SO$_2$, and in cyclic corrosion/weather resistance testing with exposure to a salt electrolyte (NaCl + (NH$_4$)$_2$SO$_4$) and UV radiation). Furthermore, the corrosion resistance of individual organic coatings was also studied using electrochemical techniques (potentiodynamic polarization studies and electrochemical impedance spectroscopy). The influence of the individual types of studied pigments on the mechanical resistance of the prepared organic coatings using five basic types of standardized mechanical tests was studied with the aim of evaluating the influence of the mechanical properties of the prepared paint film due to the presence of the individual types of tested pigments.

The results of cyclic corrosion tests in a salt mist atmosphere, in an atmosphere containing SO$_2$, in an atmosphere with salt electrolyte (NaCl + (NH$_4$)$_2$SO$_4$), and under UV radiation demonstrated the positive effect of two types of synthesized novel magnesium complex dye (Mg-Dye-I ($C_{34}H_{26}MgN_8O_6$)) and the tested inorganic pigments MgO, especially at higher values of the volume concentration of individual dyes (PVC = 5 and 10%). This conclusion is clearly confirmed by the conducted potentiodynamic polarization studies and electrochemical impedance spectroscopy, in which the above-mentioned organic coatings achieved higher corrosion resistance compared to the standard organic coating pigmented only with zinc at PVC/CPVC = 0.60. The above conclusions were additionally confirmed by the results of the determination of corrosion loss from aqueous extracts of pigments and of loose paint films. The lowest values of corrosion loss from aqueous extracts of pigments and from aqueous extracts of loose paint films (PVC = 10%) were achieved when testing the novel magnesium complex dye (Mg-Dye-I ($C_{34}H_{26}MgN_8O_6$)) and the inorganic pigment MgO. The above two types of pigments showed higher anti-corrosion performance in zinc-pigmented coatings compared to the industrially produced anti-corrosion pigment (Ca-Mg-HPO$_4$) used by paint manufacturers. Organic coatings with the above-mentioned types of pigments also achieved excellent mechanical properties, exceeding the mechanical properties of a standard organic coating pigmented only with powdered zinc. The organic coating with MgO content achieved the highest mechanical resistance compared to the other types of organic coatings tested, and its pull-off strength also reached the highest

value. The assumed mechanism of action of the studied novel magnesium complex dye in organic coatings containing zinc pigment was proposed, and the authors pointed out that the resulting mechanism was formed by a series of partial steps of the individual components of the studied organic coatings. A positive aspect of the protective action of a zinc-pigmented organic coating containing Mg pigments is the possibility of a synergistic effect of the $Zn^{2+}$ or $Mg^{2+}$ cations, which could enable a reduction in the content of metallic zinc in these types of coatings.

**Author Contributions:** Conceptualization, M.K., F.A., K.B., A.K. (Andréa Kalendová) and R.H.; Methodology, M.K., F.A., K.B., M.B., A.K. (Anna Krejčová), J.S., S.S., L.M., E.S., P.P.D. and A.A.B.; Software, M.K. and F.A.; Validation, M.K., F.A. and K.B.; Formal analysis, M.K., F.A., K.B., M.B., A.K. (Anna Krejčová), A.K. (Andréa Kalendová), P.P.D. and A.A.B.; Investigation, M.K., F.A. and K.B.; Resources, M.K., F.A. and K.B.; Data curation, M.K., F.A. and K.B.; Writing—original draft, M.K., F.A. and K.B.; Writing—review and editing, M.K., F.A., K.B., A.K. (Andréa Kalendová), R.H. and L.B.; Visualization, M.K., F.A. and K.B.; Supervision, A.K. (Andréa Kalendová), R.H. and L.B. All authors have read and agreed to the published version of the manuscript.

**Funding:** This research was funded from grants FV30048 and FV-TRIO (2016–2021) from the Ministry of Industry and Trade, Czech Republic; from GAMA2-01/002 and TG302102 from the Technology Agency of the Czech Republic; and from LM2023037 from the Ministry of Education, Youth, and Sports of the Czech Republic.

**Institutional Review Board Statement:** Not applicable.

**Informed Consent Statement:** Not applicable.

**Data Availability Statement:** Data sharing is not applicable to this article.

**Acknowledgments:** Karolína Boštíková thanks the Faculty of Chemical Technology: University of Pardubice, Czech Republic, SG 331004, for financial support. Ludmila Michalíčková thanks the Faculty of Chemical Technology, University of Pardubice, Czech Republic, SG 321003, for financial support.

**Conflicts of Interest:** The authors declare no conflict of interest.

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
