# Peer review of "New Azo Dyes-Based Mg Complex Pigments for Optimizing the Anti-Corrosion Efficiency of Zinc-Pigmented Epoxy Ester Organic Coatings"

_coatings, doi:10.3390/coatings13071276_

Round 1
Reviewer 1 Report
The manuscript is very informative and could be published. In my opinion, the material presented is extremely voluminous and more suitable to be published as a chapter of a book. On the other hand, the information presented is valuable and should be published. One of the options is to divide the text into two sub-parts, for example, or to replace the presented structures in the reactions with formulas only. I would also recommend, if the article remains as it is, that Parts 3 and 4 be somewhat revised and shortened in order to make this rather voluminous material easier to be understand.
In addition some places need corrections as presented below:
1. Abstract - PVC and PVC/CPVC to be firstly written in full and then shortened.
2. Tables 1 and 2 - "calculated" instead of "calculed".
3. Figure 14 - Magnification is not well visible.
4. Part 4.10 - In my opinion the authors should add the PDP curves or remove this section.
5. Figure 23 - Magnification is not presented.
Reviewer 2 Report
Authors examined the possibilities of using newly synthesized magnesium complex dyes in organic coatings pigmented with zinc, with the aim of reducing the zinc content in these coatings while maintaining or increasing their anti-corrosion efficiency. For the purpose of the experiment, four magnesium complexes, named Mg-Dye-I, Mg-Dye-II, Mg-Dye-III, and Mg-Dye-IV, were synthesized using a series of azocarboxylate ligands. These dyes were characterized using various analytical methods. Subsequently, model coatings containing these dyes at different concentrations, as well as coatings containing inorganic pigments, were formulated. The inhibitory corrosion properties of the individual synthesized magnesium complex dyes were investigated using standardized methods. Furthermore, the mechanical properties of the organic coatings were examined using standard tests. The research aimed to verify the potential synergistic efficacy of the new magnesium complex dyes in improving the mechanical, anti-corrosion, and chemical properties of zinc-pigmented organic coatings. In addition, the study focuses on the application of synthesized magnesium complex dyes in coatings to reduce zinc content while enhancing corrosion protection.
Before accepting this article for publication, it is necessary for the authors to resolve the listed issues:
1) The authors must carefully claim their novelty in the Section 1. Introduction. In addition, the authors need to do a some formatting errors, for example punctuation marks and citing references must be uniform, for the example, in line 58 there is dot (.) before reference, then in line 61 there is dot after reference, in line 70 there is no dot at all, and in line 101 there is dot before and after reference. The authors should uniform this item throughout the manuscript.
Also, the authors should pay the attention to the introduction of abbreviations throughout the manuscript, for the example, the abbreviation for PVC is not introduced. Authors are advised to read the manuscript carefully and correct any technical deficiencies before submitting a response to the Reviewer's concerns.
2) Within the Section 3. Experimental part, there is a content difference by subsections. Coating is an open access journal, and supports all provisions of the Open science, including Reproducibility and Replicability. In section 3.2.2. Elemental analysis the authors are advised to provide additional experimental data.
3) Line 464, 465 Table 1 and 2, Column 4, Elemental analysis (found / calculated) [%] of Oxygen should be included.
4) In the Subsection Maldi Characterization: mass spectra are necessary to be included in the manuscript to support the presented results.
5) Infrared spectroscopy (IR): Line 528 Capital word ''Figure 13'' with capital F.
Figure 13, IR spectra: the authors are advised to label important functional groups on IR spectra.
6) Line 576 ''Figure 15'' should be bold.
Line 584 ''Ca9HPO4(PO4)5OH (Calcium Hydrogen Phosphate Hydroxide)''
The authors are advised to verify the accuracy of the name of this compound, compare with this article:
Ten Huisen, K S, and P W Brown. “Formation of calcium-deficient hydroxyapatite from alpha-tricalcium phosphate.” Biomaterials vol. 19,23 (1998): 2209-17. doi:10.1016/s0142-9612(98)00131-8
7) ESI-MS results are not presented in the manuscript, although the method is mentioned in the Section Abstract, as well listed as one of the performed methods in the Section 3. Experimental part and in the Section 5. Conclusions. In the Section 4. Results and discussion this method is not even mentioned.
Minor editing
Round 2
Reviewer 2 Report
The authors have corrected manuscript in accordance wit reviewer's suggestions.